# Unifying the U–Pb and Th–Pb methods: joint isochron regression and common Pb correction

Pieter Vermeesch

London Geochronology Centre, Department of Earth Sciences, University College London, United Kingdom

**Correspondence:** Pieter Vermeesch (p.vermeesch@ucl.ac.uk)

**Abstract.** The actinide elements U and Th undergo radioactive decay to three isotopes of Pb, forming the basis of three coupled geochronometers. The $^{206}$Pb/$^{238}$U and $^{207}$Pb/$^{235}$U decay systems are routinely combined to improve accuracy. Joint consideration with the $^{208}$Pb/$^{232}$Th decay system is less common. This paper aims to change this. Co-measured $^{208}$Pb/$^{232}$Th is particularly useful for discordant samples containing variable amounts of non-radiogenic ('common') Pb.

The paper presents a maximum likelihood algorithm for joint isochron regression of the $^{206}$Pb/$^{238}$Pb, $^{207}$Pb/$^{235}$Pb, and $^{208}$Pb/$^{232}$Th chronometers. Given a set of cogenetic samples, this 'Total-Pb/U–Th algorithm' estimates the common Pb composition and concordia intercept age. U–Th–Pb data can be visualised on a conventional Wetherill or Tera-Wasserburg concordia diagram, or on a $^{208}$Pb/$^{232}$Th vs. $^{206}$Pb/$^{238}$U plot. Alternatively, the results of the new discordia regression algorithm can also be visualised as a $^{208}$Pb$_c$/$^{206}$Pb vs. $^{238}$U/$^{206}$Pb or $^{208}$Pb$_c$/$^{207}$Pb vs. $^{238}$U/$^{207}$Pb isochron, where $^{208}$Pb$_c$ represents the common $^{208}$Pb component. In its most general form, the Total-Pb/U–Th algorithm accounts for the uncertainties of all isotopic ratios involved, including the $^{232}$Th/$^{238}$U-ratio, as well as the systematic uncertainties associated with the decay constants and the $^{238}$U/$^{235}$U-ratio. However, numerical stability is greatly improved when the dependency on the $^{232}$Th/$^{238}$U-ratio uncertainty is dropped.

For detrital minerals, it is generally not safe to assume a shared common Pb composition and concordia intercept age. In this case the Total-Pb/U–Th regression method must be modified by tying it to a terrestrial Pb evolution model. Thus also detrital common Pb correction can be formulated in a maximum likelihood sense.

The new method was applied to three published datasets, including low Th/U carbonates, high Th/U allanites and overdispersed monazites. The carbonate example illustrates how the Total-Pb/U–Th method achieves a more precise common-Pb correction than a conventional $^{207}$Pb-based approach. The allanite sample shows the significant gain in both precision and accuracy that is made when the Th–Pb decay system is jointly considered with the U–Pb system. Finally the monazite example is used to illustrate how the Total-Pb/U–Th regression algorithm can be modified to include an overdispersion parameter.

All the parameters in the discordia regression method (including the age and the overdispersion parameter) are strictly positive quantities that exhibit skewed error distributions near zero. This skewness can be accounted for using the profile log-likelihood method, or by recasting the regression algorithm in terms of logarithmic quantities. Both approaches yield realistic asymmetric confidence intervals for the model parameters. The new algorithm is flexible enough that it can accommodate disequilibrium corrections and inter-sample error correlations when these are provided by the user. All the methods presented

in this paper have been added to the `IsoplotR` software package. This will hopefully encourage geochronologists to take full advantage of the entire U–Th–Pb decay system.

## 1 Introduction

The Pb content of U-bearing minerals comprises two components:

1. Non-radiogenic (a.k.a. initial or 'common') Pb is inherited from the environment during crystallisation. It contains all of lead's four stable isotopes ($^{204}$Pb, $^{206}$Pb, $^{207}$Pb and $^{208}$Pb) in fixed proportions for a given sample.

2. Radiogenic Pb is added to the common Pb after crystallisation due to the decay of U and Th. It contains only three isotopes ($^{206}$Pb, $^{207}$Pb and $^{208}$Pb), which occur in variable proportions as a function of the Th/U-ratio and age.

Denoting the measured and non-radiogenic components with subscripts '$m$' and '$c$' respectively, and assuming initial secular equilibrium, we can write:

$$^{204}Pb_m = {}^{204}Pb_c \tag{1}$$

$$^{206}Pb_m = {}^{206}Pb_c + {}^{238}U_m \left(e^{\lambda_{38}t} - 1\right) \tag{2}$$

$$^{207}Pb_m = {}^{207}Pb_c + {}^{235}U_m \left(e^{\lambda_{35}t} - 1\right) \tag{3}$$

$$^{208}Pb_m = {}^{208}Pb_c + {}^{232}Th_m \left(e^{\lambda_{32}t} - 1\right) \tag{4}$$

where $\lambda_{38}$, $\lambda_{35}$ and $\lambda_{32}$ are the decay constants of $^{238}$U, $^{235}$U and $^{232}$Th, respectively, and $t$ is the time elapsed since isotopic closure. In order to accurately estimate $t$, the common Pb composition is needed. One way to account for common Pb is to normalise all the measurements to $^{204}$Pb. For example, using the $^{238}$U – $^{206}$Pb decay scheme:

$$\left[\frac{^{206}Pb}{^{204}Pb}\right]_m = \left[\frac{^{206}Pb}{^{204}Pb}\right]_c + \left[\frac{^{238}U}{^{204}Pb}\right]_m \left(e^{\lambda_{38}t} - 1\right) \tag{5}$$

Applying Equation 5 to multiple cogenetic aliquots of the same sample defines an isochron with slope $\left(e^{\lambda_{38}t} - 1\right)$ and intercept $\left[^{206}Pb/^{204}Pb\right]_c$. Alternatively, and equivalently, an 'inverse' isochron line can be defined as:

$$\left[\frac{^{204}Pb}{^{206}Pb}\right]_m = \left[\frac{^{204}Pb}{^{206}Pb}\right]_c \left\{1 - \left[\frac{^{238}U}{^{206}Pb}\right]_m \left(e^{\lambda_{38}t} - 1\right)\right\} \tag{6}$$

In this case, the isochron is a line whose y-intercept defines the common $^{204}$Pb/$^{206}$Pb-ratio, and the x-intercept determines the radiogenic $^{238}$U/$^{206}$Pb-ratio.

The isochron concept can easily be applied to the $^{235}$U – $^{207}$Pb system, by replacing $^{206}Pb$ with $^{207}Pb$, $^{238}Pb$ with $^{235}Pb$ and $\lambda_{38}$ with $\lambda_{35}$ in Equations 5 and 6. The accuracy and precision of the calculation can be further improved by solving the $^{206}$Pb/$^{238}$U and $^{207}$Pb/$^{235}$U isochron equations simultaneously and requiring $t$ to be the same in both systems. The resulting three-dimensional constrained isochron is known as a 'Total-Pb/U isochron' (Ludwig, 1998).

In igneous samples, the conventional Total-Pb/U isochron requires isotopic data for two or more cogenetic aliquots. In the simplest case, a two-point isochron can be formed by analysing the U-Pb composition of the U-bearing phase of interest along with a cogenetic mineral devoid of U (e.g, feldspar). In detrital samples, the common Pb intercept of the isochron can be anchored to some nominal value, or to a terrestrial Pb evolution model (e.g., Stacey and Kramers, 1975). Thus, the [204]Pb-based total U-Pb isochron method is beneficial to nearly all applications of the U-Pb method.

Unfortunately, [204]Pb-based common Pb correction is not always practical. First, not all mass spectrometers are able to measure [204]Pb with sufficient precision and accuracy. In some ICP-MS instruments, the presence of an isobaric interference with [204]Hg precludes accurate [204]Pb measurements. And second, because [204]Pb is by far the least abundant of lead's four naturally occurring isotopes, it requires the longest dwell times. For single collector instruments, this reduces the precision of the other isotopes.

To overcome these problems, alternative common-Pb correction schemes have been proposed that use [207]Pb or [208]Pb instead of [204]Pb. The 'SemiTotal-Pb/U isochron' method is based on linear regression of [206]Pb–[207]Pb–[238]U-data in Tera-Wasserburg space (Ludwig, 1998; Williams, 1998; Chew et al., 2011). It assumes that all the samples are cogenetic and form a simple two component mixture between common Pb and radiogenic Pb. The common Pb then marks the intercept with the [207]Pb/[206]Pb-axis, and the radiogenic Pb can be obtained from the intersection of the isochron with the concordia line. The [207]Pb-based common Pb correction only works if the assumption of initial concordance is valid, if [207]Pb can be measured with sufficient precision, and if there is enough spread in the initial Pb/U-ratios to produce a statistically robust isochron.

Andersen (2002) introduced a [208]Pb-based common-Pb correction scheme that does not require initial concordance. His method assumes that U–Th–Pb discordance can be accounted for by a combination of Pb loss at a defined time, and the presence of common Pb of known composition. However in most cases neither the timing of Pb loss, nor the composition of the common Pb are known. Furthermore, the assumptions that underlie the Andersen (2002) method were tailored to the mineral zircon, but do not apply so much to other minerals such as carbonates, which crystallise at low temperatures and do not experience diffusive Pb loss.

This paper introduces a 'Total-Pb/U–Th isochron' algorithm that uses the $^{232}$Th – $^{208}$Pb decay scheme to determine the common Pb component. Unlike the Andersen (2002) method, it does not require the common Pb composition to be pre-specified, but assumes that no Pb-loss has occurred. The new algorithm is based on Ludwig (1998)'s Total-Pb/U isochron method, but uses $^{208}Pb_c$ instead of $^{204}Pb$ in Equation 5:

$$\frac{^{206}Pb_m}{^{208}Pb_c} = \left[\frac{^{206}Pb}{^{208}Pb}\right]_c + \frac{^{238}U_m}{^{208}Pb_c}\left(e^{\lambda_{38}t} - 1\right) \tag{7}$$

and similarly for Equation 6 and the $^{235}$U – $^{207}$Pb equivalents of Equations 5 and 6.

The algorithms introduced in this paper will be illustrated using three published U–Th–Pb datasets, which showcase how the combined U–Th–Pb approach improves both the precision and accuracy of U–Pb geochronology (Section 4). The cases studies include a carbonate dataset of Parrish et al. (2018), an allanite dataset of Janots and Rubatto (2014), and an overdispersed monazite dataset of Gibson et al. (2016). The carbonate dataset is an example of a low Th/U setting in which the [208]Pb-based common Pb correction is more precise than a conventional [207]Pb/[206]Pb-based common Pb correction. The allanite dataset is an

example of a high Th/U setting in which the [208]Pb/[232]Th method offers greater precision than the U–Pb method. The Janots and Rubatto (2014) study used SIMS and therefore also offers an opportunity to compare the new [208]Pb method with a conventional [204]Pb-based common Pb correction.

Section 5 shows how the isochron regression algorithm can be modified to accommodate strongly skewed uncertainty distributions, using a simple logarithmic change of variables. The Total-Pb/U–Th isochron algorithm assumes that all aliquots are cogenetic. However Section 6 shows how the algorithm can be adapted to detrital samples, by tying it to the two-stage Pb evolution model of Stacey and Kramers (1975). This procedure is similar in spirit to the iterative algorithm of Chew et al. (2011), but uses a maximum likelihood approach that weights the uncertainties of all isotopes in the coupled U–Th–Pb decay system. Finally, Section 7 introduces an implementation of the algorithms described herein, using the `IsoplotR` software package.

## 2 U–Th–Pb concordia and the Total-Pb/U–Th isochron

In conventional U–Pb geochronology, the set of concordant [206]Pb/[238]U- and [207]Pb/[235]U-ratios defines a 'Wetherill' concordia line. Similarly, U–Th–Pb data can be visualised in [208]Pb/[232]Th- vs. [206]Pb/[238]U-ratio space. In the absence of common Pb, samples whose [208]Pb/[232]Th-ages equal their [206]Pb/[238]U-ages plot on a U–Th–Pb concordia line. The addition of common Pb pulls samples away from this line. Binary mixing between common Pb and radiogenic Pb forms linear trends in conventional concordia space, but not in U–Th–Pb concordia space. For example the Janots and Rubatto (2014) data plot above or below the concordia line depending on the Th/U-ratio (Figure 1a).

An alternative visualisation is to plot [208]Pb/[206]Pb against [238]U/[206]Pb (Figure 1b). The radiogenic [208]Pb-component can be removed by rearranging Equation 4 for $^{208}Pb_c/^{232}Th_m$. Doing this for different values of $t$ moves the various aliquots vertically on the diagram. Each value of $t$ also corresponds to a radiogenic [238]U/[206]Pb ratio, thus marking a point on the horizontal axis of the diagram. We can fit a line through this point and minimise the residual scatter of the data around it, using a least squares criterion such as the mean square of the weighted deviates (MSWD, McIntyre et al., 1966). For the Janots and Rubatto (2014) data, the residual scatter is minimised when $t \approx 23$ Ma (Figure 1b). At this value, the aliquots plot along a simple binary mixture between common Pb and radiogenic Pb. This marks the best estimate for the concordia age. The corresponding common-Pb corrected [208]Pb/[232]Th – [206]Pb/[238]U composition is shown as a tight cluster of blue error ellipses on Figure 1a.

In order to formalise this procedure in a mathematical sense, let us first define a number of variables. In analogy to the variable names used by Ludwig (1998), we will refer to the blank corrected isotopic ratios as $X$, $Y$, $Z$, $W$ and $U$:

$$X = \left[\frac{^{207}Pb}{^{235}U}\right]_m, Y = \left[\frac{^{206}Pb}{^{238}U}\right]_m, Z = \left[\frac{^{208}Pb}{^{232}Th}\right]_m, W = \left[\frac{^{232}Th}{^{238}U}\right]_m, U = \left[\frac{^{238}U}{^{235}U}\right] \tag{8}$$

where $X$, $Y$ and $Z$ are vectors, $W$ is a diagonal matrix, and $U$ is a scalar; we will use Greek characters for the unknown common Pb ratios:

$$\alpha = \left[\frac{^{206}Pb}{^{208}Pb}\right]_c, \beta = \left[\frac{^{207}Pb}{^{208}Pb}\right]_c, \gamma = \frac{^{208}Pb_c}{^{232}Th_m} \tag{9}$$

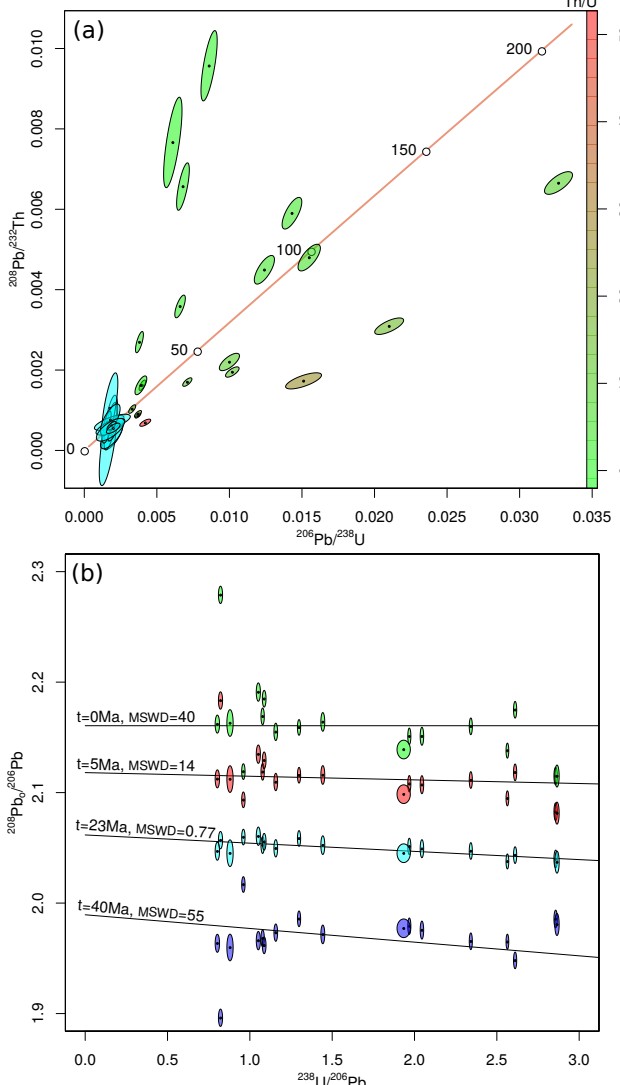

**Figure 1.** U–Th–Pb data for allanite sample MF482 of Janots and Rubatto (2014) shown on (a) a U–Th–Pb concordia diagram, and (b) a $^{208}Pb_{\circ}/^{206}Pb$ – $^{238}U/^{206}Pb$ isochron plot. The raw data are shown in shades of green to red on the concordia diagram, in proportion to the Th/U ratio. The same raw data are shown as green ellipses on the isochron diagram. Red, light and dark blue ellipses show the measurements with 5, 23 and 40 Ma worth of radiogenic $^{208}Pb$ removed, respectively. The misfit of the radiogenic $^{208}Pb$-corrected data around the best fit line is expressed as weighted square of mean deviates (MSWD, McIntyre et al., 1966) values. Error ellipses are shown at $1\sigma$.

where $\alpha$ and $\beta$ are scalars and $\gamma$ is a vector; and finally, we will use $t$ as the concordia age so that the radiogenic ratios are given by:

$$\left[\frac{^{208}Pb}{^{232}Th}\right]_* = e^{\lambda_{32}t} - 1, \quad \left[\frac{^{207}Pb}{^{235}U}\right]_* = e^{\lambda_{35}t} - 1, \quad \left[\frac{^{206}Pb}{^{238}U}\right]_* = e^{\lambda_{38}t} - 1 \tag{10}$$

Next, we define three misfit vectors $K$, $L$ and $M$ containing the difference between the measured and the predicted (i.e. common + radiogenic) isotope ratios:

$$K = X - U\beta W\gamma - e^{\lambda_{35}t} + 1 \tag{11}$$

$$L = Y - \alpha W\gamma - e^{\lambda_{38}t} + 1 \tag{12}$$

$$M = Z - \gamma - e^{\lambda_{32}t} + 1 \tag{13}$$

This formulation is a straightforward adaptation of Ludwig (1998)'s $^{204}$Pb-based Total-Pb/U isochron equations. And like Ludwig (1998), we can then estimate $t$, $\alpha$ and $\beta$ by minimising the sum of squares:

$$S = \Delta\Sigma_\Delta^{-1}\Delta^T \tag{14}$$

where $\Delta$ is the amalgamated misfit vector and $\Delta^T$ is its transpose (i.e., $\Delta^T = \begin{bmatrix} K^T & L^T & M^T \end{bmatrix}$). $\Sigma_\Delta$ is the covariance matrix of $\Delta$, which can be estimated by error propagation:

$$\Sigma_\Delta = \begin{bmatrix} J_x & J_\lambda \end{bmatrix} \begin{bmatrix} \Sigma_x & 0_{4n \times 4} \\ 0_{4 \times 4n} & \Sigma_\lambda \end{bmatrix} \begin{bmatrix} J_x^T \\ J_\lambda^T \end{bmatrix} \tag{15}$$

in which $\Sigma_x$ is the $4n \times 4n$ covariance matrix of the collated data measurements $X$, $Y$, $Z$ and $W$; $\Sigma_\lambda$ is the $4 \times 4$ covariance matrix of the decay constants and $U$, and $J_x$ and $J_\lambda$ are Jacobian matrices with partial derivatives of $\Delta$ with respect to the isotopic ratio measurements and the decay constants (plus $U$), respectively. Further details for $\Sigma_x$, $\Sigma_\lambda$, $J_x$ and $J_\lambda$ are provided in Appendix A.

Equation 14 can be solved for $t$, $\alpha$, and $\beta$ by iterative methods, but the numerical stability of these methods is not guaranteed. Numerical stability and speed of convergence can be greatly improved if we remove the uncertainties of $W$ from the data covariance matrix $\Sigma_x$. If the sum of squares $S$ does not depend on the uncertainty of $W$, then the partial derivatives of $S$ w.r.t. $\alpha$, $\beta$, $\gamma$ and $t$ can be calculated manually, which greatly simplifies the optimisation. Further details about this simplified algorithm are provided in Appendix B.

## 3  Error propagation and overdispersion

The log-likelihood of the isochron fit is given by

$$\mathcal{L} = -\frac{1}{2}\left[3n\ln(2\pi) + \ln|\Sigma_\Delta| + S\right] \tag{16}$$

where $|\Sigma_\Delta|$ marks the determinant of $\Sigma_\Delta$. The covariance matrix of the fit parameters is then obtained by inverting the matrix of second derivates of the negative log-likelihood with respect to the vector $\gamma$ and the scalars $t$, $\alpha$, $\beta$. This is also known as the

Fisher Information matrix:

$$
\begin{bmatrix}
\Sigma\gamma & s[\gamma,t] & s[\gamma,\alpha] & s[\gamma,\beta] \\
s[t,\gamma] & s[t]^2 & s[t,\alpha] & s[t,\beta] \\
s[\alpha,\gamma] & s[\alpha,t] & s[\alpha]^2 & s[\alpha,\beta] \\
s[\beta,\gamma] & s[\beta,t] & s[\beta,\alpha] & s[\beta]^2
\end{bmatrix}
= -
\begin{bmatrix}
\frac{\partial^2\mathcal{L}}{\partial\gamma^2} & \frac{\partial^2\mathcal{L}}{\partial\gamma\partial t} & \frac{\partial^2\mathcal{L}}{\partial\gamma\partial\alpha} & \frac{\partial^2\mathcal{L}}{\partial\gamma\partial\beta} \\
\frac{\partial^2\mathcal{L}}{\partial t\partial\gamma} & \frac{\partial^2\mathcal{L}}{\partial t^2} & \frac{\partial^2\mathcal{L}}{\partial t\partial\alpha} & \frac{\partial^2\mathcal{L}}{\partial t\partial\beta} \\
\frac{\partial^2\mathcal{L}}{\partial\alpha\partial\gamma} & \frac{\partial^2\mathcal{L}}{\partial\alpha\partial t} & \frac{\partial^2\mathcal{L}}{\partial\alpha^2} & \frac{\partial^2\mathcal{L}}{\partial\alpha\partial\beta} \\
\frac{\partial^2\mathcal{L}}{\partial\beta\partial\gamma} & \frac{\partial^2\mathcal{L}}{\partial\beta\partial t} & \frac{\partial^2\mathcal{L}}{\partial\beta\partial\alpha} & \frac{\partial^2\mathcal{L}}{\partial\beta^2}
\end{bmatrix}^{-1}
\tag{17}
$$

where $\Sigma_\gamma$ is an $n \times n$ matrix; $s[\gamma,t]$, $s[\gamma,\alpha]$ and $s[\gamma,\beta]$ are $n$-element row vectors, $s[t,\gamma]$, $s[\alpha,\gamma]$ and $s[\beta,\gamma]$ are $n$-element column vectors, and all other elements are scalars. The second derivatives are given in Appendix C. The Fisher Information matrix is best solved by block matrix inversion. This is achieved by partitioning Equation 17 into four parts, with $\partial^2\mathcal{L}/\partial\gamma^2$ defining the first block.

If analytical uncertainty is the only source of data scatter around the discordia line, then the sum of squares $S$ follows a central Chi-square distribution with $2n - 3$ degrees of freedom (i.e., $\chi^2_{2n-3}$). Normalising $S$ by the degrees of freedom gives rise to the so-called reduced Chi-square statistic, which is also known as the Mean Square of the Weighted Deviates (MSWD):

$$
MSWD = \frac{S}{2n-3}
\tag{18}
$$

Datasets are said to be *overdispersed* if $S$ is greater than the 95% percentile of $\chi^2_{2n-3}$ or, equivalently, if $MSWD \gg 1$ Wendt and Carl (1991). The overdispersion can either be attributed to geological scatter in the concordia intercept age $t$, or to excess variability in the common Pb ratios $\alpha$ and $\beta$. Suppose that the scatter follows a normal distribution with zero mean and let $\omega$ be the standard deviation of this distribution. Then we can redefine Equation 15 as:

$$
\Sigma_\Delta =
\begin{bmatrix} J_x & J_\lambda \end{bmatrix}
\begin{bmatrix} \Sigma_x & 0_{4n\times4} \\ 0_{4\times4n} & \Sigma_\lambda \end{bmatrix}
\begin{bmatrix} J_x^T \\ J_\lambda^T \end{bmatrix}
+ J_\omega\omega^2 J_\omega^T
\tag{19}
$$

where $J_\omega$ a the Jacobian matrix with the partial derivatives of $\Delta$ w.r.t. to the dispersion parameter $\omega$. If the overdispersion is attributed to diachronous isotopic closure, then:

$$
J_\omega =
\begin{bmatrix}
-\lambda_{35}e^{\lambda_{35}t}I_{n\times n} \\
-\lambda_{38}e^{\lambda_{38}t}I_{n\times n} \\
-\lambda_{32}e^{\lambda_{32}t}I_{n\times n}
\end{bmatrix}
\tag{20}
$$

Alternatively, if the overdispersion is attributed to excess scatter of the common Pb ratios, then:

$$
J_\omega =
\begin{bmatrix}
-UW\gamma \\
-W\gamma \\
0_{n\times1}
\end{bmatrix}
$$

$\omega$ can then be found by plugging Equation 19 into Equation 16 and maximising $\mathcal{L}$. Like before, the uncertainty of $\omega$ is obtained by inverting the Fisher Information, replacing Equation 17 with

$$
\begin{bmatrix}
\Sigma\gamma & s[\gamma,t] & s[\gamma,\alpha] & s[\gamma,\beta] & s[\gamma,\omega] \\
s[t,\gamma] & s[t]^2 & s[t,\alpha] & s[t,\beta] & s[t,\omega] \\
s[\alpha,\gamma] & s[\alpha,t] & s[\alpha]^2 & s[\alpha,\beta] & s[\alpha,\omega] \\
s[\beta,\gamma] & s[\beta,t] & s[\beta,\alpha] & s[\beta]^2 & s[\beta,\omega] \\
s[\omega,\gamma] & s[\omega,t] & s[\omega,\alpha] & s[\omega,\beta]^2 & s[\omega]^2
\end{bmatrix}
= -
\begin{bmatrix}
\frac{\partial^2 \mathcal{L}}{\partial \gamma^2} & \frac{\partial^2 \mathcal{L}}{\partial \gamma \partial t} & \frac{\partial^2 \mathcal{L}}{\partial \gamma \partial \alpha} & \frac{\partial^2 \mathcal{L}}{\partial \gamma \partial \beta} & \frac{\partial^2 \mathcal{L}}{\partial \gamma \partial \omega} \\
\frac{\partial^2 \mathcal{L}}{\partial t \partial \gamma} & \frac{\partial^2 \mathcal{L}}{\partial t^2} & \frac{\partial^2 \mathcal{L}}{\partial t \partial \alpha} & \frac{\partial^2 \mathcal{L}}{\partial t \partial \beta} & \frac{\partial^2 \mathcal{L}}{\partial t \partial \omega} \\
\frac{\partial^2 \mathcal{L}}{\partial \alpha \partial \gamma} & \frac{\partial^2 \mathcal{L}}{\partial \alpha \partial t} & \frac{\partial^2 \mathcal{L}}{\partial \alpha^2} & \frac{\partial^2 \mathcal{L}}{\partial \alpha \partial \beta} & \frac{\partial^2 \mathcal{L}}{\partial \alpha \partial \omega} \\
\frac{\partial^2 \mathcal{L}}{\partial \beta \partial \gamma} & \frac{\partial^2 \mathcal{L}}{\partial \beta \partial t} & \frac{\partial^2 \mathcal{L}}{\partial \beta \partial \alpha} & \frac{\partial^2 \mathcal{L}}{\partial \beta^2} & \frac{\partial^2 \mathcal{L}}{\partial \beta \partial \omega} \\
\frac{\partial^2 \mathcal{L}}{\partial \omega \partial \gamma} & \frac{\partial^2 \mathcal{L}}{\partial \omega \partial t} & \frac{\partial^2 \mathcal{L}}{\partial \omega \partial \alpha} & \frac{\partial^2 \mathcal{L}}{\partial \omega \partial \beta^2} & \frac{\partial^2 \mathcal{L}}{\partial \omega^2}
\end{bmatrix}^{-1}
\tag{21}
$$

In this case, manual calculation of the second derivatives is only possible if the overdispersion is attributed to $t$, with formulae shown in Appendix D. The second derivates are not tractable if the excess dispersion is assigned to $\alpha$ and $\beta$. In this case the Fisher Information must always be calculated numerically, which can be difficult.

## 4  Application to literature data

This section applies the U–Th–Pb isochron algorithm to two published datasets, a carbonate dataset of Parrish et al. (2018) and an allanite dataset of Janots and Rubatto (2014). Parrish et al. (2018) investigated the Palaeogene deformation history of southern England by dating calcite veins in chalk and greensand. The measurements were made by quadrupole LA-ICP-MS, for which it was not possible to measure $^{204}$Pb with sufficient precision or accuracy. Figure 2a shows the U–Pb data of one particular sample (CB-2, Isle of Wight) on a conventional Tera-Wasserburg diagram. In the absence of $^{204}$Pb, conventional data processing would apply a common-Pb correction using the $^{207}$Pb-method. That is, it would infer the concordia intercept age by regression of a Semitotal-Pb/U isochron. Doing so suggests a U–Pb age of $29.72 \pm 1.23$ Ma. However, this isochron exhibits significant overdispersion with respect to the analytical uncertainties (MSWD = 3.2), casting doubt on the accuracy of the date. The fit also suffers from low precision, caused by the large uncertainties of the $^{207}$Pb-measurements. These cause the error ellipses of some spots to cross over into negative $^{207}$Pb/$^{206}$Pb space.

The Th/U-ratios of CB-2 are extremely low ($<12$, as shown on the colour scale of Figure 2). These low ratios are caused by the low solubility of Th in the vein-forming fluids. As a consequence, less than 1% of the measured $^{208}$Pb is of radiogenic origin. At the same time, the sample contains between 2 and 20 times more $^{208}$Pb than it does $^{207}$Pb. This makes the $^{208}$Pb-based Total-Pb/U–Th correction far more precise than the conventional $^{207}$Pb-based Semitotal-Pb/U correction. Figure 2b shows the Total-Pb/U–Th isochron of CB-2 in $^{208}$Pb$_{\circ}$/$^{206}$Pb – $^{238}$U/$^{206}$Pb space. The scatter around this line is much tighter than that of the Semitotal-Pb/U fit, and the MSWD is only 2.5 despite the high precision of the added $^{208}$Pb data. The isochron intercept age has dropped to $24.43 \pm 0.84$ Ma, which is significantly younger than the $^{207}$Pb-corrected date. Importantly, the two age estimates do not overlap within the stated uncertainties. The new date is close to, but not quite as young as, the $22.6 \pm 1.5$ Ma value proposed by Parrish et al. (2018), which was obtained by a heuristic version of the Total-Pb/U–Th isochron algorithm.

It is not possible to formally prove that the $^{208}$Pb-corrected date is more accurate than the $^{207}$Pb-corrected date for the carbonate dataset. However, an independent assessment of accuracy *is* possible for our second case study. Janots and Rubatto (2014)'s allanite dataset used SIMS instead of LA-ICP-MS, making it possible to compare a $^{204}$Pb-based common Pb correction

with the new [208]Pb method. Figure 3a shows the U–Pb data of one particular allanite sample (MF482) on a conventional Tera-Wasserburg concordia diagram, yielding a Semitotal-Pb/U isochron age of $22.77 \pm 5.63$ Ma. This is nearly identical to Janots and Rubatto (2014)'s [207]Pb-corrected [208]Pb/[206]Pb$_\circ$ – [232]Th/[206]Pb$_\circ$ isochron age of $22.7 \pm 1.0$ Ma. As before, the Th/U-ratios are shown as shades of green to red. These values range from 23 to 235, which is three orders of magnitude higher than Parrish et al. (2018)'s carbonate data. Consequently, most of the chronometric power of the allanite data is contained in the Th–Pb system and not in the U–Pb method. 90 – 97% of the [208]Pb is radiogenic, as opposed to 0.3 – 1.0% of the [206]Pb, and only 0.06 – 0.016% of the [207]Pb.

Figure 3b shows the Th–Pb data in [204]Pb/[208]Pb – [232]Th/[208]Pb space, where they form an isochron with a Th–Pb age of $21.50 \pm 4.37$ Ma. This agrees within error with the [207]Pb-corrected U–Pb age, despite the possible presence of an unresolved isobaric interference on [204]Pb (Janots and Rubatto, 2014). However the Th–Pb isochron age has a slightly smaller uncertainty and a much lower MSWD (0.74 instead of 1.4). Combining the U–Pb and Th–Pb systems together, Figure 3c shows allanite sample MF482 in [208]Pb$_\circ$/[206]Pb – [238]U/[206]Pb space, where it defines an $23.21 \pm 0.85$ Ma isochron. This falls within the uncertainties of the U–Pb and Th–Pb age estimates, but is more than five times more precise than the previous age estimates. An alternative visualisation of the Total-Pb/U–Th isochron is shown in Figure 3d. Here, the [207]Pb$_\circ$/[208]Pb-ratio is plotted against [232]Th/[208]Pb. Thus, we use the [207]Pb as a common-Pb indicator instead of the [204]Pb used in Figure 3b. The >15 times greater abundance of [207]Pb compared to [204]Pb nearly quadruples the precision of the data, producing a tight fit around the isochron.

## 5 Dealing with skewed error distributions

All the free parameters in the regression algorithm ($t$, $\alpha$, $\beta$ and $\omega$) are strictly positive quantities. This positivity constraint manifests itself in skewed error distributions. For example, when the four parameter algorithm of Section 3 is applied to datasets that exhibit little or no overdispersion ($\omega \approx 0$), then the usual '2-sigma' error bounds can cross over into physically impossible negative data space. This section of the paper introduces two ways to deal with this problem.

A first solution is to obtain asymmetric uncertainty bounds for $\omega$ using a profile likelihood approach (Galbraith, 2005; Vermeesch, 2018). First, maximise Equation 16 for the four parameters $t$, $\alpha$, $\beta$ and $\omega$. Denote the corresponding log-likelihood value by $\mathcal{L}_m$. Second, consider a range of values for $\omega$ around the maximum likelihood estimate. For each of these values, maximise $\mathcal{L}$ for $t$, $\alpha$ and $\beta$ whilst keeping $\omega$ fixed. Denote the corresponding log-likelihood by $\mathcal{L}_\omega$. Finally, a 95% confidence region for $\omega$ is obtained by collecting all the values of $\omega$ for which $\mathcal{L}_\omega > \mathcal{L}_m - 3.85/2$, where 3.85 corresponds to the 95th percentile of a chi-square distribution with one degree of freedom (Figure 4). The same procedure can also be applied to $t$, $\alpha$ and $\beta$, in order to obtain asymmetic confidence intervals for those parameters if needed. This would be particularly useful for very young samples.

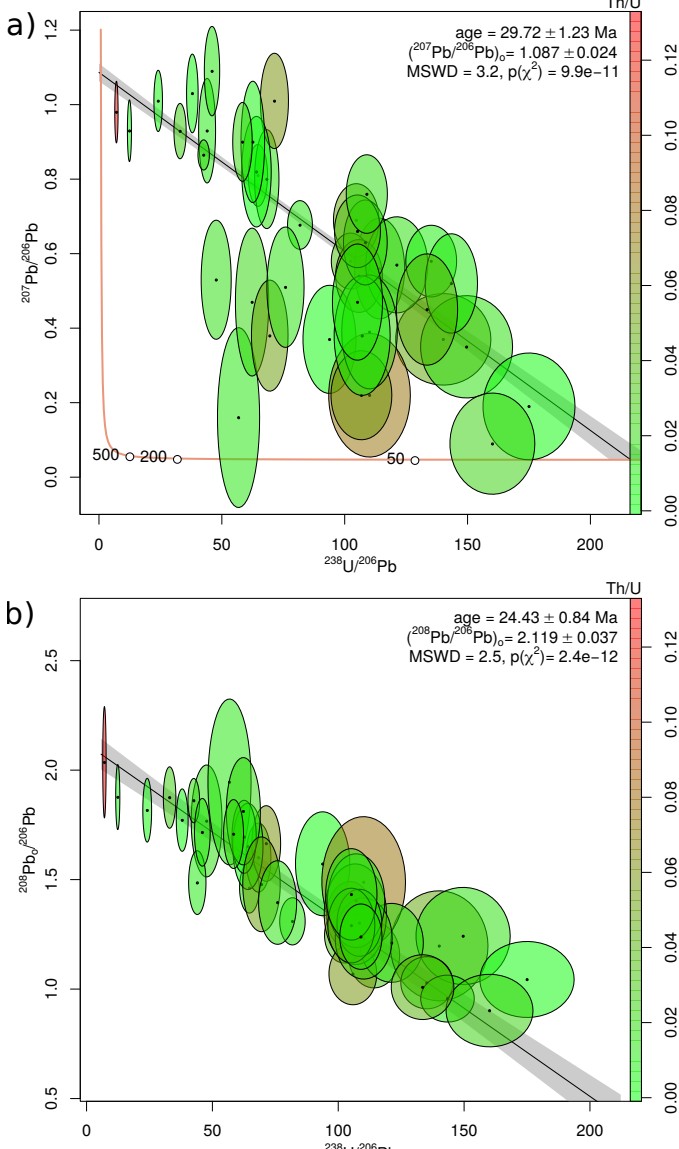

**Figure 2.** a) SemiTotal-Pb/U isochron ($^{207}$Pb-based common Pb correction) for Parrish et al. (2018)'s chalk data; b) Total-Pb/U–Th isochron ($^{208}$Pb-based common Pb correction) shown in $^{208}$Pb$_\circ$/$^{206}$Pb – $^{238}$U/$^{206}$Pb space. Colours indicate the Th/U-ratio. All uncertainties are shown at $1\sigma$.

A second and more pragmatic approach to dealing with the positivity constraint is to simply redefine the regression parameters in terms of logarithmic quantities. This is done by replacing Equations 11, 12 and 19 with:

$$K = X - U \exp[\beta_*]W\gamma - \exp[\lambda_{35}e^{t_*}] + 1 \tag{22}$$

$$L = Y - \exp[\alpha_*]W\gamma - \exp[\lambda_{38}e^{t_*}] + 1 \tag{23}$$

$$\Sigma_\Delta = \begin{bmatrix} J_x & J_\lambda \end{bmatrix} \begin{bmatrix} \Sigma_x & 0_{4n\times4} \\ 0_{4\times4n} & \Sigma_\lambda \end{bmatrix} \begin{bmatrix} J_x^T \\ J_\lambda^T \end{bmatrix} + J_\omega \exp[\omega_*]^2 J_\omega^T \qquad \textbf{10} \tag{24}$$

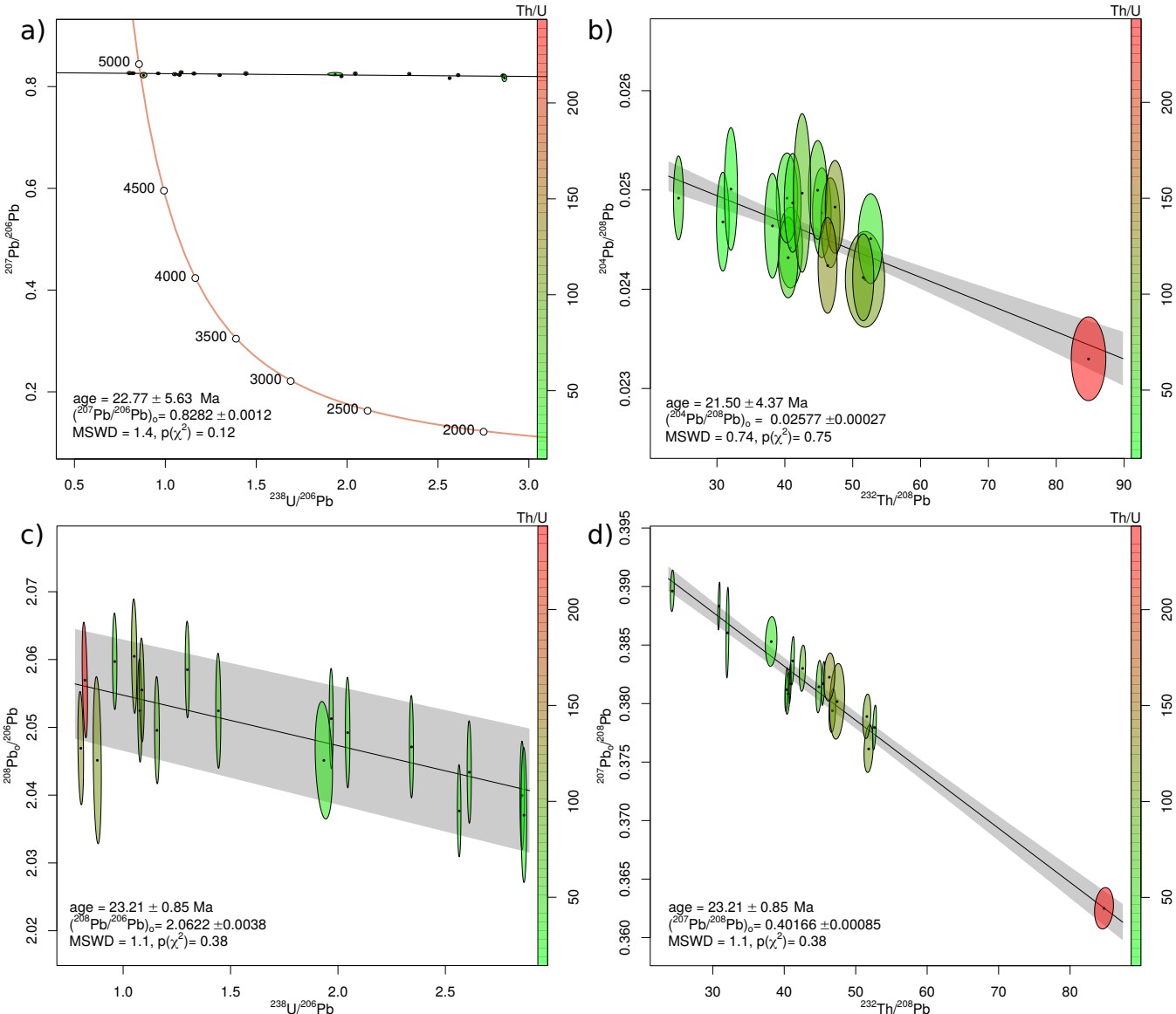

**Figure 3.** a) SemiTotal-Pb/U isochron ([207]Pb-based common Pb correction) for Janots and Rubatto (2014)'s allanite data; b) Conventional Pb/Th-isochron ([204]Pb-based common Pb correction); c) and d) Total-Pb/U–Th isochron ([208]Pb-based common Pb correction) shown in [208]Pb$_\circ$/[206]Pb – [238]U/[206]Pb space (c) and [206]Pb$_\circ$/[208]Pb – [232]Th/[208]Pb space (d). Colours indicate the Th/U-ratio. All uncertainties are shown at $1\sigma$.

respectively, and maximising Equation 16 with respect to $t_*$, $\alpha_*$, $\beta_*$ and $\omega_*$. The standard errors for these log parameters (again obtained from the Fisher Information matrix) can then be converted to asymmetric confidence intervals for $t$, $\alpha$, $\beta$ and $\omega$. This approach yields results that are similar to those obtained using the profile log-likelihood method, as illustrated in Figure 4 for

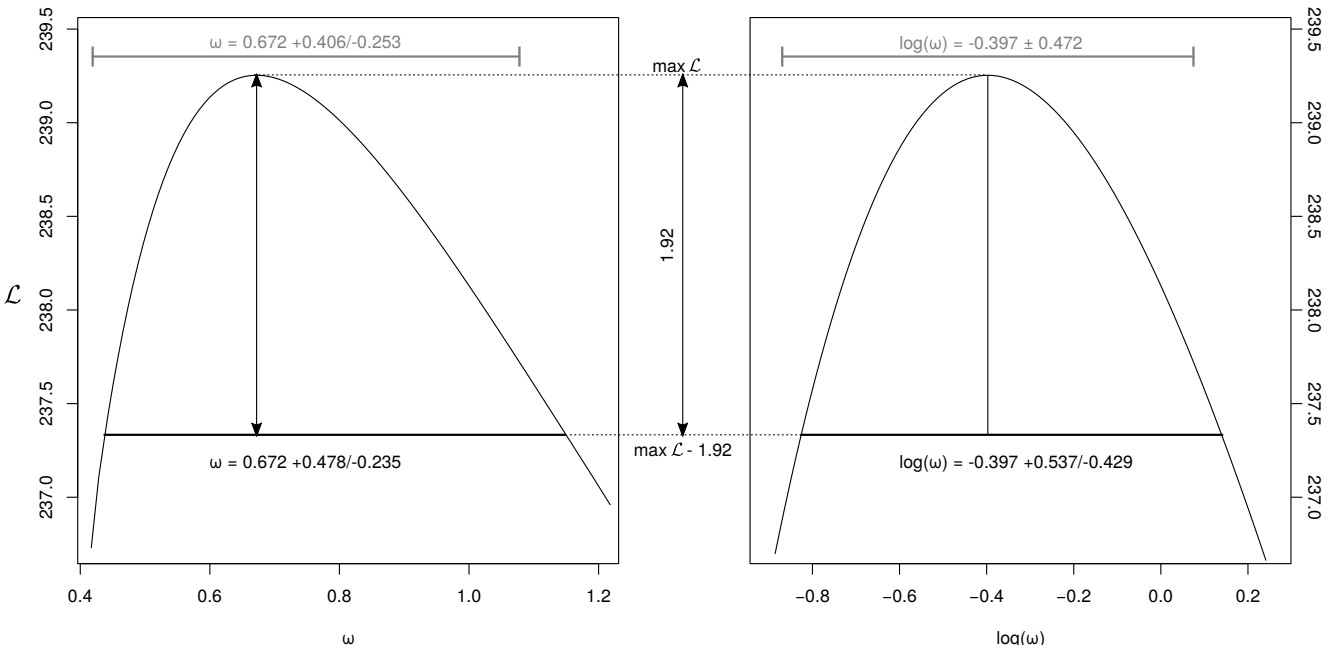

**Figure 4.** Profile log-likelihood intervals of the overdispersion parameter $\omega$ (black, left) and $\log(\omega)$ (black, right) for the Gibson et al. (2016) dataset. The set of $\omega$-values whose log-likelihood fall within a range of 1.92 from the maximum value define an asymmetric 95% confidence interval. Alternatively, a standard symmetric confidence interval for $\log(\omega)$ (grey, right) can be mapped to an asymmetric confidence interval for $\omega$ (grey, left). The two approaches yield similar results.

235 monazite grain #10 in sample BHE-01 of Gibson et al. (2016). This sample experienced a diachronous crystallisation history, resulting in an overdispersed Total-Pb/U–Th isochron fit (MSWD = 8). Quantifying the excess dispersion with a model-3 fit yields an overdispersion parameter $\omega = 0.67$ Ma with asymmetric confidence bounds of +0.48/-0.23 Ma. Besides generating realistic confidence regions, the logarithmic reparameterisation of the likelihood function has the added benefit increasing the numerical stability of the maximum likelihood method.

## 240 6 Detrital samples

So far we have assumed that all the U–Th–Pb measurements are cogenetic and share the same common Pb composition. This assumption is generally not valid for detrital minerals, which tend to contain a mixture of provenance components. In this case the different crystals in a sample are not expected to plot along a single isochron line. However it is still possible to remove the common Pb component by making certain assumptions about the common Pb composition. One way to do this is to assume

245 that the mineral of interest was extracted from a reservoir of known U–Th–Pb composition.

For example, using the two-stage Pb evolution model of Stacey and Kramers (1975), it is possible to predict the $^{206}$Pb/$^{208}$Pb and $^{207}$Pb/$^{208}$Pb ratios of the reservoir for any given time $t$. More specifically, if $t < 3.7$ Ga, then

$$\alpha(t) = \frac{\left[\frac{^{206}Pb}{^{204}Pb}\right]_{3.7} + \left[\frac{^{238}U}{^{204}Pb}\right]_{sk}\left(e^{\lambda_{38}3.7} - e^{\lambda_{38}t}\right)}{\left[\frac{^{208}Pb}{^{204}Pb}\right]_{3.7} + \left[\frac{^{232}Th}{^{204}Pb}\right]_{sk}\left(e^{\lambda_{32}3.7} - e^{\lambda_{32}t}\right)} \tag{25}$$

$$\beta(t) = \frac{\left[\frac{^{207}Pb}{^{204}Pb}\right]_{3.7} + \frac{1}{U}\left[\frac{^{238}U}{^{204}Pb}\right]_{sk}\left(e^{\lambda_{35}3.7} - e^{\lambda_{35}t}\right)}{\left[\frac{^{208}Pb}{^{204}Pb}\right]_{3.7} + \left[\frac{^{232}Th}{^{204}Pb}\right]_{sk}\left(e^{\lambda_{32}3.7} - e^{\lambda_{32}t}\right)} \tag{26}$$

where $\left[\frac{^{206}Pb}{^{204}Pb}\right]_{3.7} = 11.152$, $\left[\frac{^{208}Pb}{^{204}Pb}\right]_{3.7} = 31.23$, $\left[\frac{^{207}Pb}{^{204}Pb}\right]_{3.7} = 12.998$, $\left[\frac{^{238}U}{^{204}Pb}\right]_{sk} = 9.74$, and $\left[\frac{^{232}Th}{^{204}Pb}\right]_{sk} = 36.84$. Substituting $\alpha(t)$ and $\beta(t)$ for $\alpha$ and $\beta$ in Equations 11–13 reduces the number of free parameters from three ($\alpha$, $\beta$ and $t$) to one ($t$). This provides a quick and numerically robust mechanism for common-Pb correction of detrital minerals. It is the maximum likelihood equivalent of the heuristic approach used by Chew et al. (2011).

## 7   Implementation in `IsoplotR`

The algorithms presented in this paper have been implemented in the `IsoplotR` software toolbox for geochronology (Vermeesch, 2018). The easiest way to use the U–Th–Pb isochron functions is via an online graphical user interface at http:// isoplotr. london-geochron.com. Alternatively, the same functions can also be accessed from the command line, using the R programming language (R Core Team, 2020). This section of the paper presents some code snippets to illustrate the key functions involved. This brief tutorial assumes that the reader has R and `IsoplotR` installed on her/his computer. Further details about this are provided by Vermeesch (2018), and on the aforementioned website. First, we need to load `IsoplotR` into R:

```
library(IsoplotR)
```

Two new data formats have been added to `IsoplotR`'s existing six U–Pb formats, to accommodate datasets comprising $^{232}$Th and $^{208}$Pb. Sample Ga2 of Janots and Rubatto (2014) has been included in the `IsoplotR` package as two data files (UPb7.csv and UPb8.csv).

UPb7.csv specifies the U–Th–Pb composition using the 'Wetherill' ratios $^{207}$Pb/$^{235}$U, $^{206}$Pb/$^{238}$U, $^{208}$Pb/$^{232}$Th and $^{232}$Th/$^{238}$U, whereas UPb8.csv uses the 'Tera-Wasserburg' ratios $^{238}$U/$^{206}$Pb, $^{207}$Pb/$^{206}$Pb, $^{208}$Pb/$^{206}$Pb and $^{232}$Th/$^{238}$U. The key difference between the two formats is the strength of the internal error correlations, which is greater for format 7 than it is for format 8. The following commands load the contents of UPb8.csv into a variable called UPb, and plot the data on a $^{208}$Pb/$^{232}$Th vs. $^{206}$Pb/$^{238}$U-concordia diagram:

```
UPb <- read.data('UPb8.csv',method='U-Pb',format=8)
concordia(UPb,type=3)
```

Performing a discordia regression and visualising the results as a $^{208}$Pb$_c$/$^{206}$Pb vs. $^{238}$U/$^{206}$Pb isochron:

```
isochron(UPb,type=1)
```

which performs a three parameter regression without overdispersion. Accounting for overdispersion is done using the optional `model` argument:

```
fit <- isochron(UPb,type=1,model=3)
```

where `fit` is a variable that stores the numerical results of the isochron regression. This is a list of items that can be inspected
by typing `fit` at the R command prompt. For example, the maximum likelihood estimates for $t$, $\alpha$, $\beta$ and $\omega$ are stored in
`fit$par` and the covariance matrix in `fit$cov`. Changing `type` to 2 plots the regression results as a $^{208}Pb_c/^{207}Pb$ vs.
$^{235}U/^{207}Pb$ isochron. The isochron results can also be visualised on the concordia diagram:

```
concordia(UPb,type=2,show.age=2)
```

where `type=2` produces a Tera-Wasserburg diagram and the `show.age` argument adds a three-parameter regression line to
285 it. Change this to `show.age=4` for a four-parameter fit.

## 8   Discussion and future developments

This paper introduced a 'Total-Pb/U–Th algorithm' for common Pb correction by joint regression of all Pb isotopes of U
and Th. For samples that are low in Th (such as carbonates), $^{208}Pb$ offers the most precise way to correct for common Pb,
because $^{208}Pb$ tends to be more abundant than both $^{204}Pb$ and $^{207}Pb$. For samples that are high in Th, the $^{208}Pb/^{232}Th$ clock adds
chronometrically valuable information to the joint U–Pb decay systems.

The ingrowth of radiogenic Pb described by Equations 2–4 assumes initial secular equilibrium between all the intermediate
daughters in the U–Th–Pb decay chains. The new discordia regression algorithm can be modified to accommodate departures
from this assumption. In fact, such disequilibrium corrections have already been implemented in IsoplotR, using the matrix
derivative approach of McLean et al. (2016). A manuscript detailing these calculation is in preparation by the latter author. The
295 disequilibrium correction is particularly useful for applications to young carbonates, whose initial $^{234}U/^{238}U$ and $^{230}Th/^{238}U$
activity ratios may be far out of equilibrium.

The new discordia regression algorithm is based on the method of maximum likelihood, and accounts for correlated uncer-
tainties between variables. However geochronological datasets are often associated with equally significant error correlations
*between samples* (Vermeesch, 2015). The algorithm presented in this paper easily handles such correlations, which carry *sys-*
300 *tematic uncertainty*. These are represented by the off-diagonal terms of the covariance matrix $\Sigma_x$ in Equation 15. However, to
use this option in practical applications will require a new generation of low level data processing software.

This new generation software will also need to deal with a second issue that negatively affects the accuracy of the U(-Th)-Pb
method, which is apparent from Figure 1. After removing the radiogenic $^{208}Pb$-component from the Janots and Rubatto (2014)
dataset, the 95% confidence ellipse of one of the aliquots crosses over into negative $^{208}Pb/^{232}Th$ ratios. This nonsensical result
is related to the issues discussed in Section 5. Isotopic data are strictly positive quantities that exhibit skewed error distributions.

'Normal' statistical operations such as averaging and the calculation of '2-sigma' confidence intervals can produce counter-intuitive results when applied to such data.

In Section 5, the skewness of the fit parameters was removed by reformulating the regression algorithm in terms of logarithmic quantities. Similarly, Vermeesch (2015) showed that the skewness of isotopic compositions can be removed using log-ratios, in the context of $^{40}$Ar/$^{39}$Ar geochronology. McLean et al. (2016) introduced the same approach to in-situ U–Pb geochronology by LA-ICP-MS. Future software development will allow analysts to export their U–Th–Pb isotopic data directly as logratios and covariance matrices. Such a data structure can still be analysed with the new discordia regression algorithm, after a logarithmic change a variables for $X$, $Y$, $Z$ and $W$ in Equations 11, 12 and 13.

The U–Pb method is one of the most powerful and versatile methods in the geochronological toolbox. With two isotopes of the same parent ($^{235}$U and $^{238}$U) decaying to two different isotopes of the same daughter ($^{207}$Pb and $^{206}$Pb), the U–Pb method offers an internal quality control that is absent from most other geochronological techniques. U-bearing minerals often contain significant amounts of Th, which decays to $^{208}$Pb. However until this day geochronologists have not frequently used this additional parent-daughter pair to its full potential. It is hoped that the algorithm and software presented in this paper will change this situation.

**Appendix A**

The covariance matrix of the isotopic ratio measurements $X$, $Y$, $Z$ and $W$ is given by:

$$\Sigma_x = \begin{bmatrix} \Sigma_X & \Sigma_{X,Y} & \Sigma_{X,Z} & \Sigma_{X,W} \\ \Sigma_{Y,X} & \Sigma_Y & \Sigma_{Y,Z} & \Sigma_{Y,W} \\ \Sigma_{Z,X} & \Sigma_{Z,Y} & \Sigma_Z & \Sigma_{Z,W} \\ \Sigma_{W,X} & \Sigma_{W,Y} & \Sigma_{W,Z} & \Sigma_W \end{bmatrix} \tag{27}$$

where

$$\Sigma_X = \begin{bmatrix} s[X_1]^2 & s[X_1, X_2] & \ldots & s[X_1, X_n] \\ s[X_2, X_1] & s[X_2]^2 & \ldots & s[X_2, X_n] \\ \vdots & \vdots & \ddots & \vdots \\ s[X_n, X_1] & s[X_n, X_2] & \ldots & s[X_n]^2 \end{bmatrix}, \tag{28}$$

$$\Sigma_{X,Y} = \begin{bmatrix} s[X_1, Y_1]^2 & s[X_1, Y_2] & \ldots & s[X_1, Y_n] \\ s[X_2, Y_1] & s[X_2, Y_2] & \ldots & s[X_2, Y_n] \\ \vdots & \vdots & \ddots & \vdots \\ s[X_n, Y_1] & s[X_n, Y_2] & \ldots & s[X_n, Y_n] \end{bmatrix}, \tag{29}$$

and so forth, in which $s[a]^2$ is the variance of $a$ and $s[a,b]$ is the covariance of $a$ and $b$, for any $a$ and $b$. $\Sigma_\lambda$ is the covariance matrix of the decay constants and the $^{238}$U/$^{235}$U-ratio:

$$\Sigma_\lambda = \begin{bmatrix} s[\lambda_{35}]^2 & 0 & 0 & 0 \\ 0 & s[\lambda_{38}]^2 & 0 & 0 \\ 0 & 0 & s[\lambda_{32}]^2 & 0 \\ 0 & 0 & 0 & s[U]^2 \end{bmatrix}, \tag{30}$$

Here the covariance terms have been set to zero, but nonzero values could also be accommodated. Finally, the Jacobian matrices $J_x$ and $J_\lambda$ are given by:

$$J_x = \begin{bmatrix} I_{n \times n} & 0_{n \times n} & 0_{n \times n} & -U\beta I_{n \times n}\gamma \\ 0_{n \times n} & I_{n \times n} & 0_{n \times n} & -\alpha I_{n \times n}\gamma \\ 0_{n \times n} & 0_{n \times n} & I_{n \times n} & 0_{n \times n} \end{bmatrix} \tag{31}$$

and

$$J_\lambda = \begin{bmatrix} -t_{n \times 1}e^{\lambda_{35}t} & 0_{n \times 1} & 0_{n \times 1} & -\beta W\gamma \\ 0_{n \times 1} & -t_{n \times 1}e^{\lambda_{38}t} & 0_{n \times 1} & 0_{n \times 1} \\ 0_{n \times 1} & 0_{n \times 1} & -t_{n \times 1}e^{\lambda_{32}t} & 0_{n \times 1} \end{bmatrix} \tag{32}$$

where $0_{a \times b}$ and $1_{a \times b}$ mark a $a \times b$ matrices of zeros and ones, respectively, whereas $I_{n \times n}$ is the $n \times n$ identity matrix.

## Appendix B

The numerical stability of the optimisation is greatly enhanced by dropping the dependency of the sum of squares $S$ on the uncertainty of the Th/U ratios $W$. Thus, we replace Equation 27 by

$$\Sigma_x = \begin{bmatrix} \Sigma_X & \Sigma_{X,Y} & \Sigma_{X,Z} \\ \Sigma_{Y,X} & \Sigma_Y & \Sigma_{Y,Z} \\ \Sigma_{Z,X} & \Sigma_{Z,Y} & \Sigma_Z \end{bmatrix} \tag{33}$$

and Equation 31 by the $3n \times 3n$ identity matrix (i.e., $J_x = I_{3n \times 3n}$). Let us define $\Omega$ to be the inverse covariance matrix of $\Delta$, so that

$$\Sigma_\Delta^{-1} \equiv \Omega = \begin{bmatrix} \Omega_{1,1} & \Omega_{1,2} & \Omega_{1,3} \\ \Omega_{2,1} & \Omega_{2,2} & \Omega_{2,3} \\ \Omega_{3,1} & \Omega_{3,2} & \Omega_{3,3} \end{bmatrix} \tag{34}$$

Then, we can directly estimate $\gamma$ for any given value of $t$, $\alpha$ and $\beta$, by replacing $\gamma$ with $Z - M - e^{\lambda_{32}t} + 1$ in Equation 13, so that:

$$K = \hat{K} + U\beta WM \text{ with } \hat{K} = X - U\beta W(Z - e^{\lambda_{32}t} + 1) - e^{\lambda_{35}t} + 1 \tag{35}$$

and

$$L = \hat{L} + \alpha W M \text{ with } \hat{L} = Y - \alpha W(Z - e^{\lambda_{32} t} + 1) - e^{\lambda_{38} t} + 1 \tag{36}$$

Plugging Equations 35 and 36 into Equation 14 and rearranging yields:

$$S = M^T A M + B M + M^T C + D \tag{37}$$

where

$$A = U^2 \beta^2 W_d \Omega_{1,1} W_d + \alpha^2 W_d \Omega_{2,2} W_d + \Omega_{3,3} + U \alpha \beta W_d (\Omega_{1,2} + \Omega_{2,1}) W_d +$$
$$U \beta (W_d \Omega_{1,3} + \Omega_{3,1} W_d) + \alpha(W_d \Omega_{2,3} + \Omega_{3,2} W_d) \tag{38}$$

$$B = U \beta \hat{K}^T \Omega_{1,1} W_d + \alpha \hat{L}^T \Omega_{2,2} W_d + \alpha \hat{K}^T \Omega_{1,2} W_d + U \beta \hat{L}^T \Omega_{2,1} W_d + \hat{K}^T \Omega_{1,3} + \hat{L}^T \Omega_{2,3} \tag{39}$$

$$C = U \beta W_d \Omega_{1,1} \hat{K} + \alpha W_d \Omega_{2,2} \hat{L} + U \beta W_d \Omega_{1,2} \hat{L} + \alpha W_d \Omega_{2,1} \hat{K} + \Omega_{3,1} \hat{K} + \Omega_{3,2} \hat{L} \tag{40}$$

$$D = \hat{K}^T \Omega_{1,1} \hat{K} + \hat{K}^T \Omega_{1,2} \hat{L} + \hat{L}^T \Omega_{2,1} \hat{K} + \hat{L}^T \Omega_{2,2} \hat{L} \tag{41}$$

Taking the matrix derivative of $S$ with respect to $M$:

$$\partial S / \partial M = M^T (A + A^T) + B + C^T \tag{42}$$

Setting $\partial S / \partial M = 0$ and solving for $M$:

$$M = -(A + A^T)^{-1}(B^T + C) \tag{43}$$

Plugging $M$ back into Equation 13 yields our estimate of $\gamma$, which allows us to calculate $S$. The values of $t$, $\alpha$ and $\beta$ that minimise $S$ are then found by numerical methods.

**Appendix C**

Explicit formulae for the Fisher Information matrix (Equation 17) are possible for the simplified algorithm, in which the uncertainty of $W$ is ignored:

$$\frac{\partial^2 \mathcal{L}}{\partial \gamma^2} = - \begin{bmatrix} U \beta W \\ \alpha W \\ I_{n \times n} \end{bmatrix}^T \Sigma_\Delta^{-1} \begin{bmatrix} U \beta W \\ \alpha W \\ I_{n \times n} \end{bmatrix} \tag{44}$$

$$\frac{\partial^2 \mathcal{L}}{\partial \gamma \partial t} = \left(\frac{\partial^2 \mathcal{L}}{\partial t \partial \gamma}\right)^T = - \begin{bmatrix} U\beta W \\ \alpha W \\ I_{n\times n} \end{bmatrix}^T \Sigma_\Delta^{-1} \begin{bmatrix} (\lambda_{35}e^{\lambda_{35}t})_{n\times 1} \\ (\lambda_{38}e^{\lambda_{38}t})_{n\times 1} \\ (\lambda_{32}e^{\lambda_{32}t})_{n\times 1} \end{bmatrix} \tag{45}$$

$$\frac{\partial^2 \mathcal{L}}{\partial \gamma \partial \alpha} = \left(\frac{\partial^2 \mathcal{L}}{\partial \alpha \partial \gamma}\right)^T = \begin{bmatrix} 0_{n\times n} \\ W \\ 0_{n\times n} \end{bmatrix}^T \Sigma_\Delta^{-1} \Delta - \begin{bmatrix} U\beta W \\ \alpha W \\ I_{n\times n} \end{bmatrix}^T \Sigma_\Delta^{-1} \begin{bmatrix} 0_{n\times 1} \\ W\gamma \\ 0_{n\times 1} \end{bmatrix} \tag{46}$$

$$\frac{\partial^2 \mathcal{L}}{\partial \gamma \partial \beta} = \left(\frac{\partial^2 \mathcal{L}}{\partial \beta \partial \gamma}\right)^T = \begin{bmatrix} UW \\ 0_{n\times n} \\ 0_{n\times n} \end{bmatrix}^T \Sigma_\Delta^{-1} \Delta - \begin{bmatrix} U\beta W \\ \alpha W \\ I_{n\times n} \end{bmatrix}^T \Sigma_\Delta^{-1} \begin{bmatrix} UW\gamma \\ 0_{n\times 1} \\ 0_{n\times 1} \end{bmatrix} \tag{47}$$

$$\frac{\partial^2 \mathcal{L}}{\partial t^2} = \Delta^T \Sigma_\Delta^{-1} \begin{bmatrix} (e^{\lambda_{35}t}\lambda_{35}^2)_{n\times 1} \\ (e^{\lambda_{38}t}\lambda_{38}^2)_{n\times 1} \\ (e^{\lambda_{32}t}\lambda_{32}^2)_{n\times 1} \end{bmatrix} - \begin{bmatrix} (\lambda_{35}e^{\lambda_{35}t})_{n\times 1} \\ (\lambda_{38}e^{\lambda_{38}t})_{n\times 1} \\ (\lambda_{32}e^{\lambda_{32}t})_{n\times 1} \end{bmatrix}^T \Sigma_\Delta^{-1} \begin{bmatrix} (\lambda_{35}e^{\lambda_{35}t})_{n\times 1} \\ (\lambda_{38}e^{\lambda_{38}t})_{n\times 1} \\ (\lambda_{32}e^{\lambda_{32}t})_{n\times 1} \end{bmatrix} \tag{48}$$

$$\frac{\partial^2 \mathcal{L}}{\partial t \partial \alpha} = \frac{\partial^2 \mathcal{L}}{\partial \alpha \partial t} = - \begin{bmatrix} 0_{1\times n} \\ W\gamma \\ 0_{1\times n} \end{bmatrix}^T \Sigma_\Delta^{-1} \begin{bmatrix} (\lambda_{35}e^{\lambda_{35}t})_{n\times 1} \\ (\lambda_{38}e^{\lambda_{38}t})_{n\times 1} \\ (\lambda_{32}e^{\lambda_{32}t})_{n\times 1} \end{bmatrix} \tag{49}$$

$$\frac{\partial^2 \mathcal{L}}{\partial t \partial \beta} = \frac{\partial^2 \mathcal{L}}{\partial \beta \partial t} = - \begin{bmatrix} UW\gamma \\ 0_{n\times 1} \\ 0_{n\times 1} \end{bmatrix}^T \Sigma_\Delta^{-1} \begin{bmatrix} (\lambda_{35}e^{\lambda_{35}t})_{n\times 1} \\ (\lambda_{38}e^{\lambda_{38}t})_{n\times 1} \\ (\lambda_{32}e^{\lambda_{32}t})_{n\times 1} \end{bmatrix} \tag{50}$$

$$\frac{\partial^2 \mathcal{L}}{\partial \alpha^2} = - \begin{bmatrix} 0_{n\times 1} \\ W\gamma \\ 0_{n\times 1} \end{bmatrix}^T \Sigma_\Delta^{-1} \begin{bmatrix} 0_{n\times 1} \\ W\gamma \\ 0_{n\times 1} \end{bmatrix} \tag{51}$$

$$\frac{\partial^2 \mathcal{L}}{\partial \beta^2} = - \begin{bmatrix} UW\gamma \\ 0_{n\times 1} \\ 0_{n\times 1} \end{bmatrix}^T \Sigma_\Delta^{-1} \begin{bmatrix} UW\gamma \\ 0_{n\times 1} \\ 0_{n\times 1} \end{bmatrix} \tag{52}$$

$$\frac{\partial^2 \mathcal{L}}{\partial \alpha \partial \beta} = \frac{\partial^2 \mathcal{L}}{\partial \beta \partial \alpha} = - \begin{bmatrix} UW\gamma \\ 0_{n\times 1} \\ 0_{n\times 1} \end{bmatrix}^T \Sigma_\Delta^{-1} \begin{bmatrix} 0_{n\times 1} \\ W\gamma \\ 0_{n\times 1} \end{bmatrix} \tag{53}$$

# Appendix D

Additional derivatives are required to propagate the uncertainty of the overdispersion parameters $\omega$. This can only be done manually if the overdispersion is attributed to the concordia intercept age $t$, using the simplified model (ignoring the uncertainty of $W$). In that case

$$\frac{\partial^2 \mathcal{L}}{\partial \gamma \partial \omega} = \left( \frac{\partial^2 \mathcal{L}}{\partial \omega \partial \gamma} \right)^T = -\Delta^T \frac{\partial \Sigma_\Delta^{-1}}{\partial \omega} \begin{bmatrix} U\beta W \\ \alpha W \\ I_{n\times n} \end{bmatrix} \tag{54}$$

$$\frac{\partial^2 \mathcal{L}}{\partial t \partial \omega} = \left( \frac{\partial^2 \mathcal{L}}{\partial \omega \partial t} \right)^T = -\Delta^T \frac{\partial \Sigma_\Delta^{-1}}{\partial \omega} \begin{bmatrix} (\lambda_{35} e^{\lambda_{35} t})_{n\times 1} \\ (\lambda_{38} e^{\lambda_{38} t})_{n\times 1} \\ (\lambda_{32} e^{\lambda_{32} t})_{n\times 1} \end{bmatrix} \tag{55}$$

$$\frac{\partial^2 \mathcal{L}}{\partial \alpha \partial \omega} = \left( \frac{\partial^2 \mathcal{L}}{\partial \omega \partial \alpha} \right)^T = -\Delta^T \frac{\partial \Sigma_\Delta^{-1}}{\partial \omega} \begin{bmatrix} 0_{n\times 1} \\ W\gamma \\ 0_{n\times 1} \end{bmatrix} \tag{56}$$

$$\frac{\partial^2 \mathcal{L}}{\partial \beta \partial \omega} = \left( \frac{\partial^2 \mathcal{L}}{\partial \omega \partial \beta} \right)^T = -\Delta^T \frac{\partial \Sigma_\Delta^{-1}}{\partial \omega} \begin{bmatrix} UW\gamma \\ 0_{n\times 1} \\ 0_{n\times 1} \end{bmatrix} \tag{57}$$

$$\frac{\partial^2 \mathcal{L}}{\partial \omega^2} = -\frac{1}{2} \left( \frac{\partial^2 \ln |\Sigma_\Delta|}{\partial \omega^2} + \Delta^T \frac{\partial^2 \Sigma_\Delta^{-1}}{\partial \omega^2} \Delta \right) \tag{58}$$

with

$$\frac{\partial \Sigma_\Delta^{-1}}{\partial \omega} = -\Sigma_\Delta^{-1} \frac{\partial \Sigma_\Delta}{\partial \omega} \Sigma_\Delta^{-1} \tag{59}$$

$$\frac{\partial^2 \Sigma_\Delta^{-1}}{\partial \omega^2} = -\left( \frac{\partial \Sigma_\Delta^{-1}}{\partial \omega} \frac{\partial \Sigma_\Delta}{\partial \omega} \Sigma_\Delta^{-1} + \Sigma_\Delta^{-1} \frac{\partial^2 \Sigma_\Delta}{\partial \omega^2} \Sigma_\Delta^{-1} + \Sigma_\Delta^{-1} \frac{\partial \Sigma_\Delta}{\partial \omega} \frac{\partial \Sigma_\Delta^{-1}}{\partial \omega} \right) \tag{60}$$

$$\frac{\partial^2 \ln |\Sigma_\Delta|}{\partial \omega^2} = \mathrm{Tr} \left( \frac{\partial \Sigma_\Delta^{-1}}{\partial \omega} \frac{\partial \Sigma_\Delta}{\partial \omega} + \Sigma_\Delta^{-1} \frac{\partial^2 \Sigma_\Delta}{\partial \omega^2} \right) \tag{61}$$

in which $\mathrm{Tr}(*)$ stands for the trace of $*$ and

$$\frac{\partial \Sigma_\Delta}{\partial \omega} = 2 J_\omega^T \omega J_\omega \tag{62}$$

$$\frac{\partial^2 \Sigma_\Delta}{\partial \omega^2} = 2 J_\omega^T J_\omega \tag{63}$$

Explicit formulae for the second derivatives are not available for the common-Pb based overdispersion model. In that case, the Fisher Information matrix must be computed numerically.

*Author contributions.* As the sole author of this paper, PV did all the work and wrote the entire paper.

*Competing interests.* PV does not have any competing interests.

*Acknowledgements.* PV would like to thank Randy Parrish, Phil Hopley and John Cottle for stimulating discussions that prompted the writing of this paper; and Ryan Ickert and Kyle Samperton for careful reviews.

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
