# Peer review of "Unifying the U-Pb and Th-Pb methods: joint isochron regression and common Pb correction"

_Geochronology, 2019_

## Referee Comment (RC1) · Ryan Ickert (Referee) · 18 Nov 2019

Review of U-Th-Pb Discordia Regression by Vermeesch

This review is by Ickert (Purdue University)

This manuscript primarily describes an algorithm that can be used to determine the age of a suite of U-Th-Pb measurements (Th/U, Pb/U, and Pb isotope composition, hereafter denoted IC) that are single-stage closed systems (e.g., no Pb-loss or composite ages), have the same age, and have a range of Pb*/Pbc (ratio of radiogenic Pb to common, or initial Pb) with the same Pbc IC, without using 204Pb data. This extends what Ludwig (1998; GCA v62(4) p665-676) called the "SemiTotal-Pb/U isochron" approach for U-Pb data by adding the 232Th-208Pb system, though this link is not made explicit in the text. The algorithm is primarily described using equations, and would reach a wider audience if the plain-text explanations were expanded. Equations

for uncertainty propagation and estimation of overdispersion parameters are included. Secondarily, the manuscript describes a method to avoid non-negative compositions (section 6), using an a priori Pbc IC when analyses do not have the same age (section 7), and the implementation in IsoplotR, a software toolbox developed by the author of this manuscript.

Overall, my impression of this manuscript is that the rigor and completeness of the algorithm, and its presence in a freely available software package is a welcome addition to the literature, and that after substantial modifications I think it would be a good fit in Geochronology. For reasons discussed below, I strongly suggest that a different example dataset is used, one in which **geochronological inference is improved (relative to a conventional or published interpretation) by using this algorithm**. This could include a result that is made more precise relative to a published interpretation, or requires fewer assumptions, or is computationally less cumbersome. If such a dataset does not exist, a synthetic dataset might be appropriate, but it should be grounded in a plausible use-case. A stand-out example of this is the Ludwig (1998) paper cited in this manuscript - Figure 1 of that paper does a great job of illustrating the difficulty in chosing between any of three different ways to calculate an age and why the "Concorida Age" is a solution to the problem posed in the text. Actual Concorida Ages haven't found much of a foothold in the literature for various reasons, but the deeply narrative style of that manuscript, and the clear explanation of the systematics makes the paper a classic, even if the actual calculation is rarely used.

There are several areas of the manuscript that appear to require significant modification. I outline them briefly here and discuss in more detail below.

1) Clarity and organization. Descriptions of the technique are brief and somewhat confusing, even for an expert. Results or equations/variables are presented before they are explained, which makes reading the manuscript non-linear.

2) Despite repeated assertions that the technique "...manages to fit the data very well"

[Figure]

when applied to an example data set, it does nothing of the sort. At no point are criteria presented that describe how "success" is to be measured, but the algorithm recovers unlikely Pbc IC, a high MSWD, an age that conflicts with the original publication from which the data are derived (and is probably not correct), and implies an implausibly high Pbc concentration that could easily be checked by consulting the original analyst. To be clear, this does not mean that the technique is erroneous, but the dataset is inappropriate and violates the assumptions embedded in the technique.

3) The manuscript focusses on using the 208Pb component as an index isotope for the amount of Pbc in an analysis (rather than 204Pb), but the algorithm itself is identical to one in which either 206Pb or 207Pb are used as an index. This perhaps inadvertently demonstrates the power of this kind of rigor: the choice of index isotope or axes is irrelevant, because the algorithm comprehensively treats the full covariance structure of the data! This is not a specific problem with the manuscript per se, but it may confuse a non-specialist reader who does not realize that this approach is only a slight modification of the "SemiTotal-Pb/U isochron" approach that has been in use for decades (Ludwig 1998 and references therein).

a. This algorithm must make some effort to compare it's results to either other techniques and demonstrate that it has an advantage beyond rigorous mathematics. Many of the maths we do are approximations, and it's incumbent on the author here to demonstrate to users that there is a specific advantage to using a more complicated technique.

4) There is a rich literature on Pbc corrections in U-Pb laser ablation data and on the use of U-Pb data without 204Pb. This manuscript must engage with previous work and describe how the new algorithm fits into this well-established framework.

1) Clarity and Organization

Section 2 is confusing because it relies on results produced by the equations derived in the subsequent section. For example, the Pbc compositions (0.3685; 2.56; 11.71) and

ages (17.71 Ma) appear in this section with no context. It is never made explicit where these numbers come from – I initially thought they came from the original dataset. I infer that they are the result of the isochron algorithm described in the next section, though it's never made clear. This section would make more sense if the equations were derived first, allowing the results to be discussed in context.

Section 3 makes some similarly confusing choices. For example, the covariance matrix is introduced in equation 11, but not identified until just above equation 18 in the next column. The omegas in equation 11 are never identified. Equations 12, 13, 14 are probably the most important to make clear, but the K, L, and M are never clearly identified (described as misfit parameters above equation 18) and those equations are obscured by the use of alternative variable names. I'm sympathetic that these need to be used (e.g., X, Y, and Z, gamma, W) but the text must not make it difficult for a reader to follow and should highlight important parts for a non-specialist reader. If the author just wants to write out derivations of equations, they should be in an appendix. 12, 13 and 14 should also be written out with the original variable names ($^{206}Pb/^{238}U$, $^{207}Pb/^{235}U$ etc. ) and the significance of these equations explained to a reader.

Section 4 is just a derivation of uncertainty propagation equations. There's no effort made to provide any context or demonstrate to a reader why they should be included in the main text. As written, they simply obstruct the narrative flow. Unless there is some narrative context provided, these should be placed in an appendix.

Section 5 provides some narrative context that was lacking in Section 4, though it's not clear that all the equations are necessary for the narrative. It also provides the results of the equations in Section 4, but unfortunately spaces out all of the results, so that the age is in Section 2, the MSWD is in the beginning of section 5, the uncertainty (standard error) calculated using section 4 is at the end of section 5, along with the overdispersion parameter.

All of the results should be in one section, after the equations are derived and explained, and they should be described in context. Is this consistent with the way the samples were originally interpreted? If not, why? Why are they overdispersed? How does that influence what parameter should be used for overdispersion? Do the commonPb IC make sense?

Section 6, 7, 8 and 9 are well organized.

2) Example dataset

The example dataset is poorly described and after going through the original dataset, it's clear that it is an inappropriate dataset to use as an example here.

I'll provide the context here that is missing from the manuscript.

Gibson et al. (2016) made laser ablation ICP-MS analyses of the U-Th-Pb compositions of monazite grains that have ages that range in age from 40-15 Ma. Monazite has high concentrations of both Th and U, but usually have very high Th/U. These grains have several wt.

In Gibson et al., the $^{232}Th/^{208}Pb$ dates were used exclusively for geochronological inference because 206Pb is affected by 230Th-excess and because the 207Pb signal was very low and therefore imprecise. They interpreted the variability in the $^{232}Th/^{208}Pb$ ages as reflecting true differences in crystallization ages – the large variability in overall ages between- and within-grains, and correlations with trace element compositions strongly suggest that these variations are real.

It's hard to understand why these would be used as exemplar dataset, given the 1) the low-likelihood that individual analyses are the same age; 2) the low amount of Pbc; 3) the presence of excess 206Pb.

That the algorithm fails to recover useful geochronological information is evidenced by the results (which is a testament to the quality of the algorithm!). The Pbc compositions are inconsistent with any reasonable natural, non-radiogenic Pb. The $^{208}Pb/^{207}Pb$ value of 11.71 is implausible – values for this ratio vary but should not be more than

about 3. Similarly, the $^{208}Pb/^{206}Pb$ of 2.56 is unusually high, though not as uncommon as the $^{208}Pb/^{207}Pb$. These results furthermore imply that the grain with the highest $^{208}Pb/^{232}Th$ has approximately 20

A better explanation is the one in Gibson et al. (2016). The $^{208}Pb/^{232}Th$ vary because the analyses sampled monazite of different ages. Note that spot number 4 has a much older age (25

One additional problem is that the 238U-206Pb system suffers from excess 206Pb, which seems to have not been accounted for in the calculation in the paper. Any date that incorporates 206Pb will be biased unless this is considered – this was made clear in Gibson et al. (2016) and it is confusing as to why, in this manuscript, this is not flagged as a likely problem.

Given the facts above, it is reasonable to conclude that this example dataset does not fit the criteria needed for the algorithm described here to work. This is consistent with the results, which are overdispersed (MSWD = 8.6), inaccurate (the dates are too young), and produce physically implausible results (208/207 = 11.71).

It's hard to understand why the results are described in three places as "...it manages to fit the data very well" (line 30 p1); "...works very well for monazite" (line 73 p2) and "...the Gibson et al. (2016) test case is successful" (line 35 p7). It would be helpful for a reader if context was provided for these statements.

I strongly suggest that a different dataset is used, one in which geochronological inference is improved (relative to a conventional or published interpretation) by using this algorithm. If such a dataset does not exist, then the paper may need to be restructured substantially. An ideal case would be one in which the 204Pb is measured and available so that the effect of using it – or not – can be highlighted.

3) Using the 208 index.

I think it's useful to point out to the reader that there is nothing special about using
208Pb as the index isotope. For example, these are equations 12 13 14 from the manuscript:

$$K = \left(\tfrac{^{207}Pb}{^{235}U}\right)_m - \tfrac{^{238}U}{^{235}U} \cdot \left(\tfrac{^{207}Pb}{^{208}Pb}\right)_c \cdot \left(\tfrac{^{232}Th}{^{238}U}\right)_m \cdot \tfrac{^{208}Pb_c}{^{232}Th_m} - e^{\lambda_{235}\cdot t} + 1$$

$$L = \left(\tfrac{^{206}Pb}{^{238}U}\right)_m - \left(\tfrac{^{206}Pb}{^{208}Pb}\right)_c \cdot \left(\tfrac{^{232}Th}{^{238}U}\right)_m \cdot \tfrac{^{208}Pb_c}{^{232}Th_m} - e^{\lambda_{238}\cdot t} + 1$$

$$M = \left(\tfrac{^{208}Pb}{^{232}Th}\right)_m - \tfrac{^{208}Pb_c}{^{232}Th_m} - e^{\lambda_{232}\cdot t} + 1$$

Quantities that are to be calculated include the $\left(\tfrac{^{207}Pb}{^{208}Pb}\right)_c$, $\left(\tfrac{^{206}Pb}{^{208}Pb}\right)_c$, and $t$ (which are the same for every analysis), and 208Pbc/232m, which is different for every analysis.

What is not made clear is the similarity to, and advantage over, using the following set of two equations, derived by simply rearranging the second terms in equations 12 and 13, above

$$K = \left(\tfrac{^{207}Pb}{^{235}U}\right)_m - \tfrac{^{207}Pb_c}{^{235}U_m} - e^{\lambda_{235}\cdot t} + 1$$

or

$$K = \left(\tfrac{^{207}Pb}{^{235}U}\right)_m - \tfrac{^{206}Pb_c}{^{238}U_m} \cdot \left(\tfrac{^{207}Pb}{^{206}Pb}\right)_c - e^{\lambda_{235}\cdot t} + 1$$

$$L = \left(\tfrac{^{206}Pb}{^{238}U}\right)_m - \tfrac{^{206}Pb_c}{^{238}U_m} - e^{\lambda_{238}\cdot t} + 1$$

(This is what Ludwig (1998) called the "SemiTotal-Pb/U isochron")

In both cases (the original set of three equations and the rearranged set of two equations), for a single analysis, the system of equations is underdetermined. For two analyses, there is an exact solution, and for three or more analyses, the system is overdetermined. Rearranging the equations demonstrates that the 208Pb-index from the Th-Pb has no direct leverage on Pbc in the U-Pb system. This differs from the two U-Pb equations, which are linked via a single (207/206)c and a known 238U/235U.

It could be a useful graphical device (as used in this manuscript), but in the original equations, appears to only be a function of multiplying the middle term in K and in L by 208c/208c and 232/232. Effectively this is multiplying by 1 in order to make the equations appear interdependent.

What becomes obvious after rearrangement is that equation 14 (the 232Th-208Pb equation) is independent of the two U-Pb equations: The 208Pbc/232m only has leverage on the Th-Pb systematics, and not the U-Pb systematics. This is different than the two U-Pb chronometers, which are linked by an independently constrained and (basically) invariant 235U/238U.

I can imagine that there may be some advantage in forcing both Th-Pb and U-Pb concordance in constraining the Pbc/U, but it isn't obvious to me from this manuscript.

Given this result, it is important that the manuscript specifically describe and calculate the advantage of introducing Th-Pb data into what would otherwise be a "SemiTotal-Pb/U isochron". I recommend that the similarity between the decades-old "SemiTotal-Pb/U isochron" method and the new technique be described in more detail, as this will place the current algorithm into a proper scientific context, and give credit to previous workers, as this appears to be an advance on an established technique.

4) Previous work

This method should be placed in proper context. Including but not limited to Anderson, Chew et al., Horstwood et al.

Andersen, T., 2002. Correction of common lead in U–Pb analyses that do not report 204Pb. Chemical Geology 192, 59–79. https://doi.org/10.1016/S0009-2541(02)00195-X Chew, D.M., Petrus, J.A., Kamber, B.S., 2014. U–Pb LA–ICPMS dating using accessory mineral standards with variable common Pb. Chemical Geology 363, 185–199. https://doi.org/10.1016/j.chemgeo.2013.11.006 Horstwood, M.S., L. Foster, G., R. Parrish, R., R. Noble, S., M. Nowell, G., 2003. Common-Pb corrected in situ U–

Pb accessory mineral geochronology by LA-MC-ICP-MS. Journal of Analytical Atomic Spectrometry 18, 837–846. https://doi.org/10.1039/B304365G

Specific comments

Line 5: 232/208 is not as often considered is because there are few isotope dilution measurements of 232Th (because they are harder to make by TIMS, and few labs want to do mixed TIMS-MC-ICPMS analyses), because zircon is by far the most well used U-Th-Pb chronometer (where Th-Pb provides little additional information), and because Th/U fractionation occurs in actinide rich minerals (like allanite), complicating the systematics. The lack of statistical tools is very much a second order reason to not jointly consider all the decay schemes.

Line 7: As described above, it needs to be made clear how this advantages an analysis over, say, a SemiTotal-Pb/U Isochron. Even in the abstract, this needs to be made clear.

Line 30: It's not clear why the hyperbolic language is necessary here or on what criteria the "pinnacle of statistical rigor" is based.

Line 44-53: This is mostly true but misleading. It is possible to accurately measure 204Pb in ICPMS measurements but becomes increasingly difficult with decreasing amounts of Pbc. So Pbc-rich minerals don't necessarily suffer from this problem (and these are the minerals for which this correction is most important). This section makes it sound like a lost cause, when it is clearly not (cf. Horstwood et al, referenced above). The point about dwell time is not particularly important. Removing one isotope from a run table doesn't provide a huge improvement in on-peak time from a practical perspective (it's a square root problem), and the sentence seems to imply that efforts are made to increase count times to get high-precisions on 204Pb, which is not true. It's often a short lever, so low-precision 204Pb is perfectly adequate (percent-level precision

on 204 in ID-TIMS analyses is sufficient for «0.1

Page 5:

Section 6: This is a very important contribution and it's unfortunate that it is buried in a small section of a paper on a different topic. It's far too short to do it any justice and I hope that this receives a much more robust treatment elsewhere in the literature.

Section7: This is just a constrained Pbc regression, and it would be useful to refer to the literature where this has been done before.

Page 7:

Section 9: This is not a discussion, it is just a recap of the writing in previous sections. What would be useful, and I urge the author to do this, is to demonstrate a specific advantage of this technique (or any of those described herein) over a conventional interpretation. Show both interpretations back-to-back so we can see the advantage. This technique is certainly more sophisticated than what has come before, but if it doesn't enhance our understanding of the world around us by materially improving the way we make geochronological inferences from data, then it is just complexity for the sake of complexity. I urge the author to make an affirmative demonstration that this technique has genuine utility.

1. Does the paper address relevant scientific questions within the scope of GChron? yes

2. Does the paper present novel concepts, ideas, tools, or data? yes

3. Are substantial conclusions reached? no

4. Are the scientific methods and assumptions valid and clearly outlined? Valid but could be more clear

5. Are the results sufficient to support the interpretations and conclusions? no, because the example does a poor job of illustrating the algorithm

6. Is the description of experiments and calculations sufficiently complete and precise to allow their reproduction by fellow scientists (traceability of results)? yes

7. Do the authors give proper credit to related work and clearly indicate their own new/original contribution? no but this is an easy fix

8. Does the title clearly reflect the contents of the paper? No, it is very general

9. Does the abstract provide a concise and complete summary? yes

10. Is the overall presentation well structured and clear? no but this is straightforward to fix

11. Is the language fluent and precise? yes

12. Are mathematical formulae, symbols, abbreviations, and units correctly defined and used? Not everything is clearly defined

13. Should any parts of the paper (text, formulae, figures, tables) be clarified, reduced, combined, or eliminated? The equations that have no narrative value (e.g., section 4 and 5) should probably be separated into an appendix.

14. Are the number and quality of references appropriate? No, reference to more previous literature would be appropriate.

15. Is the amount and quality of supplementary material appropriate? yes

---

## Referee Comment (RC2) · Kyle Samperton (Referee) · 20 Feb 2020

The manuscript entitled "U-Th-Pb discordia regression" by P. Vermeesch describes a novel algorithm (implemented in the R statistical programming environment) that integrates the U-238 and U-235 U-Pb geochronometers with the relatively underutilized U-Th decay system. The algorithm is incorporated into the author's previously published IsoplotR package, permitting the calculation of isochrons and the generation of publication-quality graphics with ease. The paper is succinctly written and technically sound, and represents a valuable contribution to the U-Th-Pb literature. My comments are relatively minor in nature and, upon being addressed, I recommend publication of the manuscript in Geochronology.

Comments:

Page 1, Lines 3-4: "The 206Pb/238U and 207Pb/235U decay systems are routinely

combined to improve accuracy". May be more appropriate to have something along the lines of "...are routinely combined to improve the assessment of accuracy"?

Page 1, Lines 28-31: "Nevertheless, it manages to fit the data very well. The method should work even better for low-Th phases such as carbonates." These sentences are far too subjective and informal, please rewrite.

Page 1, Lines 62-62: "...variable proportions as a function of the Th/U-ratio and age." Technically, the proportions are a function of the Th/U-ratio, age, AND the 238U/235U-ratio. Here and later in the manuscript the author assumes the mean terrestrial zircon 238U/235U value (137.818, without uncertainty) of Hiess et al. (2012). While for many (most?) applications of the algorithm this assumption is possibly acceptable, insofar as broadening the general applicability of this approach I think it is worth stating this point explicitly. For example, the algorithm may find use in early Solar System/cosmochemical studies, in which the applicability of the Hiess et al. mean terrestrial zircon U isotopic composition is suspect, and U isotopic variability often observed and thereby requiring direct 238U/235U measurement.

Page 1, Lines 64-65 thru Page 2, Lines 1-4 (equations 1-4) As presented, equations 1-4 consist of only measured (m) and non-radiogenic, initial crystallization (c) Pb components, and assume only initial secular equilibrium. However, technically the equations also assume a trivial blank Pb component during analysis/sample loading. For example, equation 1 equates the measured Pb-204 with the non-radiogenic, initial Pb-204. This is only true in practice is the Pb-204 contribution from blank is trivial, which may not necessarily be the case (especially in the case of Pb!). Here and throughout the manuscript a quantitative blank correction is not addressed, which is fine, but if so a statement should be made here that the equations as currently formulated assume a trivial Pb blank component.

Page 2, Lines 71-74: "The fact that the new algorithm works very well for monazite implies that it is generally applicable low Th phases such as carbonates." First, there is

a "to" missing between "low" and "Th". More importantly, however, I suggest you show as oppose to tell on this point. You mention in the abstract, here, and in the concluding discussion that this approach is especially useful for carbonates. Couldn't you pull a representative carbonates dataset to demonstrate this point explicitly? I'd be interested to see this.

Page 3, Line 25 (fixed 238U/235U-ratio): See my comment above re. the U isotopic composition.

Page 5, Lines 16-17 (MSWD discussion): You mention in passing that data are "overdispersed if...MSWD »1". However, I think it worth stating a more general point about the acceptable MSWD range as a function of the number of degrees of freedom (i.e., data points), a la Wendt and Carl (1991). I think it worth citing Wendt and Carl (1991) here, as well as presenting a general formula for the range/uncertainty on the MSWD itself, beyond stating the oversimplification that data are overdispersed when MSWD»1.

Page 6, Line 77 (discussion of accessing algorithm in R): You should cite R for those not in the know, the most recent suggested citation I give below.

Relevant references: 1. Wendt, I. and C. Carl, 1991. The statistical distribution of the mean squared weighted deviation. Chemical Geology (Isotope Geoscience Section) 86(4):275–285, doi: 10.1016/0168-9622(91)90010-T. 2. R Core Team, 2017. R: A Language and Environment for Statistical Computing. R Foundation for Statistical Computing, Vienna, Austria. URL https://www.R-project.org/

[Figure]

---

## Author Comment (AC1) · 9 Mar 2020

I am grateful to Dr. Ickert for his review, which is one of the most careful and detailed ones that I have ever received. The text raises a number of pertinent points, which I will address in the revised manuscript. Following the format of the review, I will first give a general response to the most important points, and this will be followed by a detailed response to the specific comments.

1. Clarity and organisation

   Ludwig (1998)'s "Treatment of concordant U/Pb ages" is one of my favourite papers of all time, because it is concise yet provides sufficient mathematical detail to verify the derivations and translate the algorithm into computer code. It was my aim to give my manuscript those same two qualities. However it appears that I

have taken the concision too far in some places, whilst providing too much mathematical detail elsewhere. I will expand some of the descriptive text and move some of the mathematical detail to an appendix.

2. Example data

The reviewer points out that the reanalysis of Gibson et al. (2016)'s monazite U-Pb data is "at odds with the published results" due to a combination of true age heterogeneity and initial $^{230}$Th/$^{238}$U-disequilibrium. The example data used in the manuscript was taken from one specific low-Y monazite crystal (grain #10) in one specific sample (BHE-01). The reported $^{208}$Pb/$^{232}$Th-ages ages within this particular grain are fairly uniform, with a weighted mean of 19.9±0.2 Ma. This is significantly older than the U-Th-Pb isochron age (17.8±0.3 Ma). It is unlikely that the difference is due to initial $^{230}$Th/$^{238}$U-disequilibrium, because correcting for this would move the age into the wrong direction. Repeating the $^{208}$Pb/$^{232}$Th-age calculations of Gibson et al. (2016) shows that these authors did not apply a common Pb correction to their data. So I have good reasons to believe that the U-Th-Pb isochron age is in fact more accurate than the published values.

The reviewer is correct that the common Pb intercepts are too high. These estimates are imprecise, and the high MSWD reflects the difficulty of the U-Th-Pb isochron algorithm to fit both the U and Th data. So I will follow Dr. Ickert's suggestion and replace this example with two new ones: a carbonate dataset of Parrish et al. (2018) and an allanite dataset of Janots and Rubatto (2014). The carbonate dataset is an example of a low Th/U setting in which the $^{208}$Pb-based common Pb correction is more precise than a conventional $^{207}$Pb/$^{206}$Pb-based common Pb correction (see Figure 1 of this response letter). The allanite dataset is an example of a high Th/U setting in which the $^{208}$Pb/$^{232}$Th method offers greater precision than the U-Pb method. The Janots and Rubatto (2014) study

used SIMS and so it is also possible to compare a [204]Pb-based common Pb correction with the new [208]Pb method. The comparison is favourable to the new U-Th-Pb isochron algorithm (Figure 2 of the response letter).

3. Novelty

Dr. Ickert writes that the isochron method presented in my manuscript "is only a slight modification of [Ludwig's] 'SemiTotal-Pb/U isochron' approach." and that the "advantage in forcing both Th-Pb and U-Pb concordance in constraining the Pbc/U [...] isn't obvious to [him] from this manuscript." First, the new algorithm is not based on Ludwig (1998)'s **Semi**Total-Pb/U isochron method, but on his **Total**-Pb/U method. Second, the two new datasets will better illustrate the power of including Th-Pb in the isochron analysis. In the case of low-Th/U carbonate data, I will cite the relevant section of Parrish et al. (2018):

> "This approach allows common [206]Pb to be quantified more robustly than methods using either [204]Pb or [207]Pb because the [208]Pbc can be determined more precisely than using [204]Pb, [207]Pb or a combination of the two. In samples with low Th/U ratio this approach has two major advantages: (1) uncertainties of individual analyses are smaller, resulting in less scatter and improved uncertainty of isochron arrays; (2) it allows more reliable calculation of single spot ages and their weighted means. For most analyses, the uncertainties in measurement and consequent estimation of common Pb are smaller for [208]Pb/[206]Pb than for [207]Pb/[206]Pb. In all cases in this study, for spots with >60% radiogenic Pb, both regression ages agree within uncertainty. In all samples the ages and uncertainties of [U-Th-Pb isochron] regressions and weighted means of [208]Pb-corrected single spot ages agree within uncertainty, and both generally have smaller uncertainties and less regression scatter than analogous [207]Pb-corrected methods."

For high-Th/U phosphate data, most of the geochronological power lies in the $^{208}Pb/^{232}Th$ clock. This chronometer lacks the equivalent of the U-Pb clock's $^{207}Pb/^{206}Pb$-based common-Pb correction. In the absence of $^{204}Pb$, the newly developed U-Th-Pb isochron is the only way to account for common Pb. I will add these details to the paper.

4. References

The original manuscript did not cite existing common Pb correction schemes proposed by Andersen (2002), Horstwood et al. (2003), Chew et al. (2014) among others. I will add these references to the revised manuscript, whilst highlighting their underlying assumptions and limitations. More specifically, the method of Andersen (2002) assumes that U-Th-Pb discordance "can be accounted for by a combination of lead loss at a defined time, and the presence of common lead of known composition". This is clearly not the case for the carbonate and allanite data discussed in the revised manuscript; the $^{204}Pb$-based approach of Horstwood et al. (2003) is complicated in the presence of $^{204}Hg$ and is imprecise due to the low abundance of $^{204}Pb$ (see Figure 2.b); and the limitations of $^{207}Pb$-based methods as discussed by Chew et al. (2014) have already been explained in the quote by Parrish et al. (2018) given above.

**Response to the detailed comments**

The reviewer was puzzled why

"the Pbc compositions (0.3685; 2.56; 11.71) and ages (17.71 Ma) appear in [Section 2] with no context."

[Figure]

The optimal common Pb composition and age could be obtained by trial and error, until the samples plot along a line in Pb/Pb–U/Pb space. To clarify this point, I will add some truly random guesses for the concordia age to the plot. See Figure 3 of this response letter. Please note that this new figure uses the Janots and Rubatto (2014) data instead of the Gibson et al. (2016) data from the original manuscript.

> "the covariance matrix is introduced in equation 11, but not identified until just above equation 18 in the next column."

Equation 11 contains five different parameters, which are defined in terms of other parameters. Explaining the meaning of all these parameters takes space. I will address this issue by moving lines 110-120 to an Appendix.

> "The omegas in equation 11 are never identified."

Here I simply followed Ludwig (1998): the omegas are defined implicitly in terms of the inverted covariance matrix.

> "If the author just wants to write out derivations of equations, they should be in an appendix. 12, 13 and 14 should also be written out with the original variable names ($^{206}$Pb/$^{238}$U, $^{207}$Pb/$^{235}$U etc.) and the significance of these equations explained to a reader."

It is not easy to fit the original variables in *GChron*'s two-column format. But what I can do is follow Ludwig (1998) and define the variables before instead of after using them. Equations 18-20, 32-41 and 46-55 will be moved to an appendix.

> "there is nothing special about using $^{208}$Pb as the index isotope"

[208]Pb was chosen as an index isotope so as to replace [204]Pb in Ludwig (1998)'s Total-Pb/U algorithm. This is different from the alternative formulations proposed by the reviewer, which refer to the SemiTotal-Pb/U algorithm. It is true that the Total-Pb/U regression problem can be redefined in terms of Tera-Wasserburg variables instead of the current Wetherill variables. But the solution is easier and cleaner in Wetherill space.

> "Line 5: 232/208 is not as often considered because there are few isotope dilution measurements of [232]Th (because they are harder to make by TIMS, and few labs want to do mixed TIMS-MC-ICPMS analyses), because zircon is by far the most well used U-Th-Pb chronometer (where Th-Pb provides little additional information), and because Th/U fractionation occurs in actinide rich minerals (like allanite), complicating the systematics. The lack of statistical tools is very much a second order reason to not jointly consider all the decay schemes."

[208]Pb and [232]Th are easy to measure by LA-ICP-MS, which has become by far the most widely used analytical technique for U-Th-Pb geochronology. Zircon is indeed the most widely used mineral phase for U-Pb geochronology, but in recent years there has been a rapid rise in the number of studies that use other mineral phases such as apatite, allanite, rutile, and carbonates. Two examples of such studies will be included in the revised manuscript, showcasing the gains in accuracy and precision that can be made with the U-Th-Pb isochron method. The effects of Th/U fraction can quite easily be quantified by comparing the Th/U ratio of the dated mineral with that of the whole rock (Schärer, 1984). This correction has already been implemented in `IsoplotR`.

> "It is possible to accurately measure [204]Pb in ICPMS measurements but becomes increasingly difficult with decreasing amounts of Pbc. So Pbcrich minerals don't necessarily suffer from this problem (and these are the minerals for which this correction is most important)."

Speaking from experience, I am unable to accurately measure $^{204}$Pb using my quadrupole LA-ICP-MS instrument at UCL, even with gold filters. The blank is more than 90% of the signal. For young and U,Th-poor samples, it is difficult enough to measure the radiogenic Pb, let alone the common $^{204}$Pb.

> "The point about dwell time is not particularly important. Removing one isotope from a run table doesn't provide a huge improvement in on-peak time from a practical perspective (it's a square root problem)"

In the case of Janots and Rubatto (2014)'s allanite study, there is 38 times more $^{208}$Pbc than $^{204}$Pb (Figure 2.b). So for the same dwell time, the $^{208}$Pbc measurement would be more than six times more precise than the $^{204}$Pb measurement. Conversely, the same precision can be achieved for $^{208}$Pbc in one sixth of the time as $^{204}$Pb. Conclusion: the square root problem is important.

> "Section 6: This is a very important contribution and it's unfortunate that it is buried in a small section of a paper on a different topic. It's far too short to do it any justice and I hope that this receives a much more robust treatment elsewhere in the literature."

By moving much of the mathematical detail to an appendix, Section 6 will gain prominence. My solution to the problem of asymmetic confidence intervals will be further explored in a forthcoming paper on disequilibrium corrections that I will co-author with Dr. Noah McLean and others later this year.

"Section 7: This is just a constrained Pbc regression, and it would be useful
to refer to the literature where this has been done before."

I will add another reference to Chew et al. (2014) here.

"What would be useful, and I urge the author to do this, is to demonstrate
a specific advantage of this technique (or any of those described herein)
over a conventional interpretation. Show both interpretations back-to-back
so we can see the advantage."

See Figures 1 and 2 of this response letter, which will be added to the revised
manuscript.

"8. Does the title clearly reflect the contents of the paper? No, it is very
general"

The title of Ludwig (1998) is also very general ("On the treatment of concordant
uranium-lead ages"). But I will follow the reviewer's suggestion and change the title
to: "Unifying the U–Pb and Th–Pb methods: joint isochron regression and common
lead correction".

**References**

Andersen, T. Correction of common lead in U–Pb analyses that do not report $^{204}$Pb. *Chemical
geology*, 192(1-2):59–79, 2002.

Chew, D., Petrus, J., and Kamber, B. U–Pb LA–ICPMS dating using accessory mineral stan-
dards with variable common Pb. *Chemical Geology*, 363:185–199, 2014.

Gibson, R., Godin, L., Kellett, D. A., Cottle, J. M., and Archibald, D. Diachronous deformation along the base of the Himalayan metamorphic core, west-central Nepal. *Geological Society of America Bulletin*, 128(5-6):860–878, 2016.

Horstwood, M. S., Foster, G. L., Parrish, R. R., Noble, S. R., and Nowell, G. M. Common-Pb corrected in situ U–Pb accessory mineral geochronology by LA-MC-ICP-MS. *Journal of Analytical Atomic Spectrometry*, 18(8):837–846, 2003.

Janots, E. and Rubatto, D. U–Th–Pb dating of collision in the external Alpine domains (Urseren zone, Switzerland) using low temperature allanite and monazite. *Lithos*, 184:155–166, 2014.

Ludwig, K. R. On the treatment of concordant uranium-lead ages. *Geochimica et Cosmochimica Acta*, 62:665–676, 1998. doi: 10.1016/S0016-7037(98)00059-3.

McIntyre, G. A., Brooks, C., Compston, W., and Turek, A. The Statistical Assessment of Rb-Sr Isochrons. *Journal of Geophysical Research*, 71:5459–5468, 1966.

Parrish, R. R., Parrish, C. M., and Lasalle, S. Vein calcite dating reveals Pyrenean orogen as cause of Paleogene deformation in southern England. *Journal of the Geological Society*, 175(3):425–442, 2018.

Schärer, U. The effect of initial $^{230}$Th disequilibrium on young UPb ages: the Makalu case, Himalaya. *Earth and Planetary Science Letters*, 67(2):191–204, 1984.
* * *
**Fig. 1.** a) SemiTotal-Pb/U isochron (207Pb-based common Pb correction) for Parrish et al. (2018)'s chalk data; b) Total-Pb/U-Th isochron (208Pb-based common Pb correction).

**Fig. 2.** a) SemiTotal-Pb/U isochron for Janots and Rubatto (2014)'s allanite data; b) Conventional Pb/Th-isochron; c) and d) Total-Pb/U-Th isochron.

**Fig. 3.** U-Th-Pb data for chalk samples CB-2 of Parrish et al. (2018) shown on a U-Th-Pb concordia diagram. Colours indicate the Th/U-ratio. All uncertainties are shown at 1 sigma.

---

## Author Comment (AC2) · 13 Mar 2020

I would like to thank Dr. Samperton for his positive review. Most of his comments are easy to address apart from one remark about the $^{238}U/^{235}U$-ratio, which prompted me to confront a bigger issue that I had avoided in the original manuscript.

> Page 1, Lines 3-4: "The $^{206}Pb/^{238}U$ and $^{207}Pb/^{235}U$ decay systems are routinely combined to improve accuracy". May be more appropriate to have something along the lines of "...are routinely combined to improve the assessment of accuracy"?

It is true that, in many geological applications, the $^{238}Pb–^{206}Pb$ and $^{235}Pb–^{207}Pb$ clocks are simply plotted together to assess concordance, after which a simple weighted mean $^{206}Pb/^{238}U$ age is calculated. However, Ludwig (1998) showed that the clocks can

also be combined to estimate a hybrid (concordia or isochron) age, which in theory is more accurate than either the $^{206}$Pb/$^{238}$U or $^{207}$Pb/$^{235}$U age. The aim of the U-Th-Pb isochron paper is to explore this application further but including the $^{208}$Pb/$^{232}$Th clock as well. So in this case I maintain that "improving acccuracy" is a more appropriate term than "improving the assessment of accuracy".

> Page 1, Lines 28-31: "Nevertheless, it manages to fit the data very well. The method should work even better for low-Th phases such as carbonates." These sentences are far too subjective and informal, please rewrite.

I will add two new datasets to the paper, including a carbonate example (Parrish et al., 2018) and an allanite example (Janots and Rubatto, 2014). By comparing conventional common-Pb corrections for these data with the new Total-Pb/U-Th algorithm, the improvement in precision and accuracy will be much clearer to the reader. See Figures 1 and 2 of this response letter for further details.

> Technically, the proportions are a function of the Th/U-ratio, age, AND the $^{238}$U/$^{235}$U ratio. Here and later in the manuscript the author assumes the mean terrestrial zircon $^{238}$U/$^{235}$U value (137.818, without uncertainty) of Hiess et al. (2012). While for many (most?) applications of the algorithm this assumption is possibly acceptable, insofar as broadening the general applicability of this approach I think it is worth stating this point explicitly.

The reviewer is correct that the $^{238}$U/$^{235}$U ratio affects the $^{207}$Pb/$^{206}$Pb ratio. However as long as all the analyses are cogenetic (which is a requirement for isochron regression), departure of the $^{238}$U/$^{235}$U ratio from the Hiess et al. (2012) values actually does not hurt the accuracy of the isochron age. This is because, in Equation 12 of the original manuscript, $^{238}$U/$^{235}$U is multiplied with the common-Pb ratio $\beta$. So as long as

$^{238}$U/$^{235}$U and $\beta$ do not vary between aliquots, an overestimation of one translates into an underestimation of the other without affecting $t$.

So the uncertainty of the $^{238}$U/$^{235}$U ratio ($U$ in Equation 12) only matters for the error propagation of $\beta$. It is not easy to address this issue with the maximum likelihood formulation of the original manuscript, in which $U$ occurs in a product with $\gamma$. If the uncertainty of $U$ is to be propagated, it is no longer possible to reformulate the sum of squares $S$ in terms of the Th/Pb misfit parameter $M$ (Equation 23). Similarly, the analytical uncertainty of the measured $^{232}$Th/$^{238}$U ratio ($W$ in Equations 12-14) is also difficult to propagate.

The solution to both of these problems is straightforward in theory, but complicated in practice. Recalling the general equation for the sum-of-squares (Equation 11 of the original manuscript):

$$S = \Delta^T \left( J^T \Sigma J \right)^{-1} \Delta$$

we can replace Equations 12 (for $J$) and 13 (for $\Sigma$) with

$$\Sigma = \begin{bmatrix} s[X]^2 & s[X,Y] & s[X,Z] & s[X,W] & 0_{n\times 1} & 0_{n\times 1} & 0_{n\times 1} & 0_{n\times 1} \\ s[Y,X] & s[Y]^2 & s[Y,Z] & s[Y,W] & 0_{n\times 1} & 0_{n\times 1} & 0_{n\times 1} & 0_{n\times 1} \\ s[Z,X] & s[Z,Y] & s[Z]^2 & s[Z,W] & 0_{n\times 1} & 0_{n\times 1} & 0_{n\times 1} & 0_{n\times 1} \\ s[W,X] & s[W,Y] & s[W,Z] & s[W]^2 & 0_{n\times 1} & 0_{n\times 1} & 0_{n\times 1} & 0_{n\times 1} \\ 0_{1\times n} & 0_{1\times n} & 0_{1\times n} & 0_{1\times n} & s[\lambda_{35}]^2 & 0 & 0 & 0 \\ 0_{1\times n} & 0_{1\times n} & 0_{1\times n} & 0_{1\times n} & 0 & s[\lambda_{38}]^2 & 0 & 0 \\ 0_{1\times n} & 0_{1\times n} & 0_{1\times n} & 0_{1\times n} & 0 & 0 & s[\lambda_{32}]^2 & 0 \\ 0_{1\times n} & 0_{1\times n} & 0_{1\times n} & 0_{1\times n} & 0 & 0 & 0 & s[U]^2 \end{bmatrix}$$

and

$$
J = \begin{bmatrix}
1_{n,n} & 0_{n\times n} & 0_{n\times n} \\
0_{n\times n} & 1_{n\times n} & 0_{n\times n} \\
0_{n\times n} & 0_{n\times n} & 1_{n\times n} \\
-U\beta\gamma & -\alpha\gamma & 0_{n\times n} \\
-t_{1\times n}e^{\lambda_{35}t} & 0_{1\times n} & 0_{1\times n} \\
0_{1\times n} & -t_{1\times n}e^{\lambda_{38}t} & 0_{1\times n} \\
0_{1\times n} & 0_{1\times n} & -t_{1\times n}e^{\lambda_{32}t} \\
-\beta W\gamma & 0_{1\times n} & 0_{1\times n}
\end{bmatrix}
$$

respectively. Unfortunately, taking matrix derivatives of $S$ is difficult to do by hand for this generalised formulation. In well behaved cases, R's optimisation function manages to calculate them numerically. But the numerical stability of these solutions is significantly poorer than that of the original algorithm.

An additional advantage of the new formulation is its ability to accommodate a second type of overdispersion model. Section 5 of the original manuscript parameterised the overdispersion in terms of the concordia intercept age. With the generalised formulation of the maximum likelihood problem, it is also possible to attribute the excess dispersion to the common Pb composition. In this case we replace Equation 44 of the original manuscript with the following alternative:

$$
J_\omega = \begin{bmatrix}
-UW\gamma \\
-W\gamma \\
0_{n\times n}
\end{bmatrix}
$$

Again, the numerical stability of this formulation is not as good as that of the original algorithm. If I find a way to increase this stability, then I will use the new algorithm. Otherwise I will stick with the original version and be more clear about its limitations.

throughout the manuscript a quantitative blank correction is not addressed, which is fine, but if so a statement should be made here that the equations as currently formulated assume a trivial Pb blank component.

I will add a line to clarify that the data are assumed to have been blank corrected.

Couldn't you pull a representative carbonates dataset to demonstrate this point explicitly? I'd be interested to see this.

A carbonate example will be added to the revised manuscript. See Figure 1 of this response letter.

You mention in passing that data are "overdispersed if... MSWD ≫1". However, I think it worth stating a more general point about the acceptable MSWD range as a function of the number of degrees of freedom (i.e., data points), a la Wendt and Carl (1991). I think it worth citing Wendt and Carl (1991) here, as well as presenting a general formula for the range/uncertainty on the MSWD itself, beyond stating the oversimplification that data are overdispersed when MSWDÂż1.

I will add a reference to Wendt and Carl (1991). However it is also important not to overly rely on MSWDs and p-values. It is possible for a precise dataset with an MSWD value of 100 to be more valuable than an imprecise dataset with an MSWD of 1. What matters is not so much whether a dataset is overdispersed or not, but rather *how* dispersed it is. This key point is addressed in Section 5 of the paper.

You should cite R for those not in the know

I will add the requested citation.

**References**

Hiess, J., Condon, D. J., McLean, N., and Noble, S. R. $^{238}$U/$^{235}$U systematics in terrestrial uranium-bearing minerals. *Science*, 335(6076):1610–1614, 2012.

Janots, E. and Rubatto, D. U–Th–Pb dating of collision in the external Alpine domains (Urseren zone, Switzerland) using low temperature allanite and monazite. *Lithos*, 184:155–166, 2014.

Ludwig, K. R. On the treatment of concordant uranium-lead ages. *Geochimica et Cosmochimica Acta*, 62:665–676, 1998. doi: 10.1016/S0016-7037(98)00059-3.

Parrish, R. R., Parrish, C. M., and Lasalle, S. Vein calcite dating reveals Pyrenean orogen as cause of Paleogene deformation in southern England. *Journal of the Geological Society*, 175(3):425–442, 2018.

[Figure]

a)

age = 29.72 ± 1.23 Ma
$(^{207}Pb/^{206}Pb)_o$= 1.087 ±0.024
MSWD = 3.2, p($\chi^2$) = 9.9e−11

$^{207}Pb/^{206}Pb$

Th/U

500 ○ 200 ○                    50 ○

$^{238}U/^{206}Pb$

b)

age = 24.43 ± 0.84 Ma
$(^{208}Pb/^{206}Pb)_o$= 2.119 ± 0.037
MSWD = 2.5, p($\chi^2$)= 2.4e−12

$^{208}Pb_o/^{206}Pb$

Th/U

$^{238}U/^{206}Pb$

**Fig. 1.** a) SemiTotal-Pb/U isochron (conventional 207Pb-based common Pb correction) for Parrish et al. (2018)'s chalk data; b) Total-Pb/U-Th isochron (new 208Pb-based common Pb correction).

**Fig. 2.** a) Conventional SemiTotal-Pb/U isochron for Janots and Rubatto (2014)'s allanite data;
b) 204Pb based Pb/Th-isochron; c) and d) new Total-Pb/U-Th isochron.

---

## Editor Comment (EC1) · Brenhin Keller (Editor) · 13 Apr 2020

Thank you for the thorough responses to reviewers. I find that the revised manuscript addresses the reviewer comments well to the extent of my expertise. Due to the technical nature of the manuscript and of the revisions requested by Dr. Ickert, I have asked Dr. Ickert to comment further on the revised manuscript. I trust that the final proof will reflect your consideration of these comments, enclosed below.

Brenhin Keller

[Figure]

**GChronD**

Interactive
comment

I would like to thank Dr. Vermeesch for responding positively to my long review, and also for the additions and that he has made to the manuscript – in my opinion this has made a good manuscript even stronger. At the request of the AE I was asked to look at the new case studies, and I have three brief comments.

1) These are excellent case studies and include both high and low $^{208}$Pb*/$^{208}$Pbc. Mixed U-Th-Pb studies are rare, and Dr. Vermeesch is to be commended for finding good exemplars. 2) Unfortunately, there is a problem with the case study on allanite that may require that the Pb/U comparison results (not the results from the new algorithm) be adjusted. In ion microprobe analyses of REE-rich minerals like allanite and monazite, there is an unresolvable molecular interference near mass station 204 that complicates the measurement of $^{204}$Pb. This is mentioned briefly in Janots and Rubatto (2014) near the top of page 158 and explains why the original authors did not utilize $^{204}$Pb in any of their data analysis or interpretations. This isobar is described elsewhere in the literature by e.g. Gregory et al. (2007; 10.1016/j.chemgeo.2007.07.029) and Stern and Berman (2000; 10.1016/S0009-2541(00)00239-4) but is probably not well known outside of the ion microprobe community. My understanding is that it is likely to be a complex REE (possibly Nd?) molecule, but I am not aware of any published literature that can attest to this. It is easy to miss the short reference to this in the original manuscript. I am not intimately familiar with the original dataset, but I would think that if the original authors deem the $^{204}$Pb data too suspect to be utilized, it is probably not suitable for an example dataset here. Adjusting the comparison to include only U, Th, and the radiogenic Pb isotopes (similar to the Parrish et al. dataset) should be straightforward and will not detract from the quality of the manuscript. 3) I would strongly suggest that the ages interpreted by the original authors be included for comparison. I understand and agree that an "apples-to-apples" comparison is useful and necessary in the context of this manuscript (in this case, a comparison of the new
algorithm with a semi-total Pb/U isochron), but out of fairness to the original authors, their own interpretation should be compared as well. While this is *clearly not the intention of the author*, it is easy for a reader to misunderstand that the Pb/U ages in the manuscript (which are imprecise and may be inaccurate) are the interpretations presented in Parrish et al. and Janots and Rubatto. In reality, both original papers recognized that the best course of action is to combine all of the ($^{204}$Pb-free) U-Th-Pb data – in a broadly similar, but less rigorous, way than the present manuscript. The Parrish paper in particular is predicated on the utility of leveraging Th-Pb data along with U-Pb, when $^{204}$Pb cannot be reliably measured, in order to improve the accuracy and precision of U-Th-Pb dates. Both Parrish et al. and Janots and Rubatto arrived at dates that are similar to those determined using the new algorithm. In both cases, the new dates are superior, both in rigour and in precision, so this takes nothing away from the utility of the new algorithm.

(signed)

Ryan Ickert
* * *

---

## Author Response (AR2)

**Response to the remaining comments on gchron-2019-14**

Pieter Vermeesch
UCL Earth Sciences
p.vermeesch@ucl.ac.uk

1. I thank Dr. Ickert for spotting the reference to the possible unresolved isobaric interference on $^{204}$Pb in the Janots and Rubatto (2014) dataset. I have decided *not* to remove the conventional Th–Pb isochron from the paper for the following reasons. First, the main reason why Janots and Rubatto (2014) did not use a $^{204}$Pb-based common Pb correction is the low abundance of this isotope in their sample, and *not* the alleged isobaric interference. Second, the proposed isobaric interference is only mentioned as a possibility and not as a fact. As Dr. Ickert pointed out, the first paper to report the possibility of an interference on $^{204}$Pb in monazite was Stern and Berman (2001). However in monazite this interference is associated with a correlation between $ThO_2$ content and apparent $^{204}$Pb counts. No such correlation is observed in allanite and so the presence of the interference is quite speculative. Third, applying the $^{204}$Pb-based common Pb correction to the Janots and Rubatto (2014) dataset actually yields results that are in excellent agreement with the $^{207}$Pb-based and $^{208}$Pb-based alternatives. To me this suggests that the $^{204}$Pb data are actually fine to use. However I have added a sentence to the revised manuscript about the possible interference.

2. As requested, the revised manuscript reports the published ages for the Parrish et al. (2018) and Janots and Rubatto (2014) datasets. As the reviewer remarkes, these are consistent with the new results, but are less precise and less robust.

3. I have changed all instances of the words 'lead' and 'uranium' to their chemical symbols 'Pb' and 'U'.

**References**

Janots, E. and Rubatto, D. U–Th–Pb dating of collision in the external Alpine domains (Urseren zone, Switzerland) using low temperature allanite and monazite. *Lithos*, 184:155–166, 2014.

Parrish, R. R., Parrish, C. M., and Lasalle, S. Vein calcite dating reveals Pyrenean orogen as cause of Paleogene deformation in southern England. *Journal of the Geological Society*, 175(3):425–442, 2018.

Stern, R. A. and Berman, R. G. Monazite U–Pb and Th–Pb geochronology by ion microprobe, with an application to in situ dating of an Archean metasedimentary rock. *Chemical Geology*, 172(1-2):113–130, 2001.